# Effects of a Novel Infant Formula on the Fecal Microbiota in the First Six Months of Life: The INNOVA 2020 Study

**DOI:** 10.3390/ijms24033034

**Published:** 2023-02-03

**Authors:** Francisco Javier Ruiz-Ojeda, Julio Plaza-Diaz, Javier Morales, Guillermo Álvarez-Calatayud, Eric Climent, Ángela Silva, Juan F. Martinez-Blanch, María Enrique, Marta Tortajada, Daniel Ramon, Beatriz Alvarez, Empar Chenoll, Ángel Gil

**Affiliations:** 1Department of Biochemistry and Molecular Biology II, School of Pharmacy, University of Granada, 18071 Granada, Spain; 2Instituto de Investigación Biosanitaria IBS.GRANADA, Complejo Hospitalario Universitario de Granada, 18014 Granada, Spain; 3RG Adipocytes and Metabolism, Institute for Diabetes and Obesity, Helmholtz Diabetes Center at Helmholtz Center Munich, Neuherberg, 85764 Munich, Germany; 4Institute of Nutrition and Food Technology “José Mataix”, Centre of Biomedical Research, University of Granada, Avda. del Conocimiento s/n., 18016 Armilla, Spain; 5Children’s Hospital of Eastern Ontario Research Institute, Ottawa, ON K1H 8L1, Canada; 6Product Development Department, Alter Farmacia SA, 28880 Madrid, Spain; 7Gregorio Marañón Maternal and Children’s Hospital, 28009 Madrid, Spain; 8ADM-BIOPOLIS, Scientific Park Universitat de València, 46980 Paterna, Spain; 9CIBEROBN, CIBER Physiopathology of Obesity and Nutrition, Instituto de Salud Carlos III, 28029 Madrid, Spain

**Keywords:** arachidonic acid, α-lactalbumin, *Bifidobacterium animalis* subsp. *lactis*, body composition, docosahexaenoic acid, postbiotics

## Abstract

Exclusive breastfeeding is highly recommended for infants for at least the first six months of life. However, for some mothers, it may be difficult or even impossible to do so. This can lead to disturbances in the gut microbiota, which in turn may be related to a higher incidence of acute infectious diseases. Here, we aimed to evaluate whether a novel starting formula versus a standard formula provides a gut microbiota composition more similar to that of breastfed infants in the first 6 months of life. Two hundred and ten infants (70/group) were enrolled in the study and completed the intervention until 12 months of age. For the intervention period, infants were divided into three groups: Group 1 received formula 1 (INN) with a lower amount of protein, a proportion of casein to whey protein ratio of about 70/30 by increasing the content of α-lactalbumin, and with double the amount of docosahexaenoic acid/arachidonic acid than the standard formula; INN also contained a thermally inactivated postbiotic (*Bifidobacterium animalis* subsp. *lactis*). Group 2 received the standard formula (STD) and the third group was exclusively breastfed (BF) for exploratory analysis. During the study, visits were made at 21 days, 2, 4, and 6 months of age, with ±3 days for the visit at 21 days of age, ±1 week for the visit at 2 months, and ±2 weeks for the others. Here, we reveal how consuming the INN formula promotes a similar gut microbiota composition to those infants that were breastfed in terms of richness and diversity, genera, such as *Bacteroides, Bifidobacterium*, *Clostridium*, and *Lactobacillus*, and calprotectin and short-chain fatty acid levels at 21 days, 2 and 6 months. Furthermore, we observed that the major bacteria metabolic pathways were more alike between the INN formula and BF groups compared to the STD formula group. Therefore, we assume that consumption of the novel INN formula might improve gut microbiota composition, promoting a healthier intestinal microbiota more similar to that of an infant who receives exclusively human milk.

## 1. Introduction

Exclusive breastfeeding is highly recommended for the first six months of life [1] because it promotes adequate growth and development, excellent nutritional status, and reduces infant morbidity and mortality in both emerging [2] and industrialized countries [3,4]. Remarkably, human milk is a biological system with interacting components that affect both the mother and the child, improving their health [5,6]. Thus, human milk offers many nutrients, especially bioactive and immunogenic substances, which support not only infant growth and development, but also the maturity of the immune system. This supports gut protection and maturation [7]; however, from the age of 6 months, children should start eating safe and adequate complementary foods.

The gut microbiota is essential in maintaining or restoring human health from early in life [8], and breastfeeding could be protective against dysbiosis [9]. However, formula-fed infants exhibit significant changes in the intestinal microbiota, which have been related to a higher incidence of infectious diseases compared to those exclusively breastfed [10]. Nevertheless, whenever breastfeeding is difficult or even impossible for some mothers, formula milk can be used to supplement the infant’s nutritional needs, which has been intended to mimic human milk by adding bioactive ingredients such as postbiotics, among others [11], while continuing to breastfeed for up to 2 years [12]. Continuous research is being conducted on infant formulas to improve their composition by incorporating new food ingredients and bioactive compounds that contribute to the child’s optimal development and functionality [13].

Infant formulas’ protein content is generally higher than that of human milk and offers all essential amino acids in sufficient amounts [14]. Although these factors promote growth and weight gain, they also increase the risk of obesity and metabolic diseases in adulthood [15]. Indeed, a lower amount of protein intake should be tested to ensure the content is as close to human milk as possible. The relatively high levels of long-chain polyunsaturated fatty acids of both the n-6 and n-3 series, especially arachidonic acid (AA, 20:4 n-6) and docosahexaenoic acid (DHA, 22:6 n-3), in human milk has led to the incorporation of these nutrients in infant formula in recent years. Therefore, the formula should provide DHA at 0.3–0.5 % of total fatty acids and a minimal amount of AA equivalent to the DHA content, and this supplementation should be clinically tested [15,16].

On the other hand, research on the use of prebiotics, probiotics, synbiotics, postbiotics, parabiotics, and paraprobiotics in infant formulas has arisen in recent years. In particular, the probiotic strain *Bifidobacterium animalis* subsp. *lactis* reduces fat mass in the visceral adipose tissue of individuals with obesity [17,18,19], and its inactivated form (postbiotic) has been shown to modulate the gut microbiota composition [20]. Furthermore, studies have shown that increasing the use of probiotics can help to prevent several chronic infant diseases, such as necrotizing enterocolitis and atopic eczema, and improve short- and long-term health [21,22]. The use of synbiotics leads to changes in the gut microbiota composition as well. At this time, the clinical outcomes of the supplementation of probiotics in infant formulas need to be evaluated, and future studies need to be assessed the long-term effects.

We hypothesized that children fed in the first 6 months of life with a novel starting infant formula (INN), compared to those fed a standard formula (STD), should develop a microbiota as similar as possible to the microbiome developed in breastfed children (control or BF). The INN formula contains a lower amount of protein, a proportion of casein to whey protein ratio of about 70/30 by increasing the content of α-lactalbumin, and with double the amount of DHA/AA than the standard formula; it also contains a postbiotic, a thermally inactivated bacteria (*Bifidobacterium animalis* subsp. *lactis*). Therefore, we evaluated whether the novel starting formula against a standard formula provides a gut microbiota composition similar to that of breastfed infants in the first 6 months of life.

## 2. Results

### 2.1. Phylum Level

In this study, we observed that both richness and diversity were similar between the INN and STD groups, and both cases presented higher values than the BF group at 21 days. At 2 and 6 months, the INN and BF groups exhibited similar values without significant differences, while STD displayed the highest values. At the phylum level, the relative abundance was similar to groups at 21 days, 2 months, and 6 months. However, *Proteobacteria* showed a decrease in the relative abundance per treatment (*p* = 0.008) and per visit between BF and both formulas INN and STD (*p* < 0.001) (Table 1 and Appendix A). The most abundant phylum was *Actinobacteria*, followed by *Firmicutes*, *Verrucomicrobia*, *Proteobacteria*, and *Bacteroides*. Appendix A shows the fold change of absolute counts between visit 4 at 6 months compared to visit 1 at 21 days at the most abundant phylum. Compared to the BF group and INN formula, the interaction time × treatment from the STD group increased in the Shannon index (*p* < 0.001), inverse Simpson (*p* = 0.004), Pielou’s evenness (*p* < 0.001), and Simpson (*p* < 0.001) at 21 days, 2 months, and 6 months. Furthermore, the Fisher index (*p* = 0.015) and species richness (*p* = 0.023) were different per visit, with the highest relative abundance at 6 months (Table 1).

### 2.2. Genus Level

At the genus level, the relative abundance of *Bifidobacterium* exhibited differences per treatment (*p* = 0.002), per visit (*p* < 0.001), and in the interaction time × treatment (*p* < 0.001), at 21 days, 2 and 6 months, with a higher abundance in the BF and INN groups compared to the STD group. Moreover, the *Bacteroides* group also exhibited differences per visit (*p* = 0.023), with the relative abundance of INN being more similar to the BF group (Table 2), although we observed a lower fold change of absolute counts between visit 1 at 21 days and visit 4 at 6 months in the BF group compared to INN (Appendix A). However, we observed the opposite effects for *Clostridium sensu stricto 1*, where the STD group presented the highest relative abundance compared to the INN and BDF groups at 2 months (*p* = 0.039). On the other hand, we observed differences either per treatment or per visit. *Collinsella* showed differences per visit (*p* < 0.001), and *Akkermansia* had differences per treatment (*p* < 0.001). *Streptococcus* showed differences per treatment and visit, with lower values in the STD group (*p* < 0.001) (Table 2).

At 21 days, we observed differences in absolute counts between the INN and STD groups. Thus, the INN group exhibited lower levels of *Blautia*, while higher levels of *Clostridium* were shown. At 2 months, the major genus continued to be *Bifidobacterium*, followed by *Pseudoescherichia*, and then *Bacteroides* and *Veillonella*. Here, we highlight the genera of *Bacteroides, Parabacteroides*, *Erysipelatoclostridium*, and *Clostridium* as having lower levels in the BF group, followed by infants fed with INN, and as having higher levels in those fed with the STD. On the contrary, the genera *Lactobacillus*, *Staphylococcus*, *Streptococcus*, and *Bifidobacterium* presented higher absolute counts in children fed with BF, followed by INN and STD. Appendix A shows the main genera expressed as fold change of absolute counts between visit 1 at 21 days and visit 4 at 6 months, where it is exhibited how the STD group differs from the INN and BF groups.

### 2.3. Species Levels

At 2 months, *Bifidobacterium breve* was found in greater counts in the BF group, followed by the INN group, while *B. longum* was higher in both formula groups compared with the BF group. The species *L. paracasei*, *S. aureus*, and *S. salivarius* presented higher counts in the BF group, followed by INN and with different levels to the STD. The levels of *C. difficile* were lower in the BF and INN groups compared to the STD. At 6 months, mainly bifidobacteria, with the species *B. longum* and *B. breve*, exhibited the greatest presence and dominated the gut microbiota profile. However, unlike what happened at 2 months of age, in this case no significant differences were obtained among groups, but levels of *B. bifidum* were greatest in absolute counts in the BF group, followed by the INN formula. We should note that *Ruminococcus gnavus*, *Akkermansia muciniphila*, and *C. difficile* exhibited lower levels in the BF and INN groups compared to infants fed the STD formula (Appendix A).

### 2.4. Rivera-Pinto Microbiome Balance

As a result of using the Rivera-Pinto balance method for microbiome analyses [23], it has been identified that the *Firmicutes* and *Actinobacteria* phyla, as well as the *Anaerostipes*, *Lactobacillus*, and UBA1819 genera, were most associated with the BF group when comparing the INN formula group with the BF group (Figure 1A). Concerning BF group samples, higher balance scores were associated with larger relative abundances of *Firmicutes* and *Actinobacteria* phyla, and *Anaerostipes*, *Lactobacillus*, and UBA1819 genera when compared to *Proteobacteria* phylum and *Veillonella*, *Flavonifractor*, and *Ruminococcus torques* group genera (Figure 1A).

*Lactobacillus* genus was most associated with the BF group in comparison to the STD group. Higher balance scores were associated with elevated relative abundances of *Lactobacillus* in BF group samples (Figure 1B) for *Veillonella*, *Flavonifractor*, *Ruminococcus torques* and *gnavus* groups, *Akkermansia*, *Bifidobacterium*, and *Anaerostipes* genera. The AUC of 0.719 indicates moderate discrimination accuracy between the STD and BF groups (Figure 1B).

With regard to the INN and STD groups, *Ruminococcus gnavus* group, *Akkermansia* genera, *Proteobacteria*, and *Bifidobacterium* were most associated with the STD group when comparing the INN formula group with the STD formula group (Figure 1C). Thus, the AUC of 0.686 indicates a poor discrimination accuracy between the INN and STD groups (Figure 1C).

### 2.5. IgA, Calprotectin and Short-Chain Fatty Acids (SCFAs)

The fecal-secreted IgA values were higher for the BF group at 21 days compared to the two formula-fed infant groups. Similarly, at 2 months, the IgA values in the BF group remained at similar levels to those of 21 days and were significantly increased in the children fed with both formulas. However, no differences were found between the INN and STD groups. At 6 months, IgA values decreased in the BF group, being still significantly higher compared to STD, while the INN group exhibited values closer to those of the BF group (Figure 2A). In the case of fecal calprotectin levels, we did not note differences between INN and BF at 21 days, 2 months, and 6 months (Figure 2B).

The fecal-secreted IgA values were higher for the BF group for the entire duration of the study, compared to the two formula-fed infant groups. However, at two months of life, the INN and STD groups showed a significant increase in IgA (*p*-value < 0.0001), indistinguishable between the two groups, and without reaching BF levels. At 6 months, IgA levels decreased in all groups, most notably in the STD group (*p*-value < 0.0001) (Figure 2A). In the case of fecal calprotectin levels, we did not note differences between the three groups at 21 days and 2 months. However, at 6 months, calprotectin levels decreased for the INN and BF groups, remaining high at the STD group (Figure 2B).

We also analyzed the SCFAs and lactic acid in fecal samples and we detected that lactic acid was higher in the BF and INN groups compared to the STD at 2 and 6 months of life, being statistically significant at 6 months (Figure 3A). For acetic acid, no significant differences were found between INN and STD; however, the BF group showed higher levels for the entire duration of the study, being significant at 2 months (Figure 3B). In the case of propionic acid, the BF group presented the lowest values throughout the study. We only found differences between the INN and STD groups at 21 days, and no differences were shown between infants fed with either the INN or STD formulas at 2 and 6 months. However, for the STD group, the propionic acid content was significantly higher compared to the BF group at 6 months (Figure 3C). For the INN and STD groups, butyrate levels tended to be higher at 21 days, with STD levels also higher at 2 months and 6 months, while the BF group showed very low values throughout the study. In general, the INN group was more similar to the BF than the STD group. At 6 months, butyrate levels in the STD group were significantly higher compared to the BF group (Figure 3D).

### 2.6. Correlations between Bacterial Diversity Indices, Bacterial Variables, SCFAs Levels, Metabolic Traits, and Clinical Outcomes

Pearson’s correlations between bacterial diversity indices, bacterial variables, SCFAs levels, metabolic traits, and clinical outcomes revealed that there were some associations related to the BF, STD, and INN groups (Appendix A).

*Firmicutes* was inversely correlated with *Actinobacteria* and positively correlated with secreted IgA levels in the INN formula group, at a statistically significant level. The Shannon index and propionic acid were negatively associated with *Bifidobacterium*, while lactic acid was positively associated with IgA. The Shannon index was positively correlated with *Collinsella*, *Anaerostipes*, *Ruminococcus torques* group, *Faecalibacterium*, *Flavoniflactor*, and *Clostridium sensu stricto* 1. UBA1819, *Flavoniflactor*, and *Akkermansia* were positively associated with L-tryptophan and dTDP-N-acetylthomosamine biosynthesis, while *Veillonella* was negatively associated (Appendix A). A positive correlation was found between bronchiolitis and *Faecalibacterium*, but also with calprotectin levels, in the STD group at six months. *Bifidobacterium* showed a negative correlation with L-tryptophan, dTDP-N-acetylthomosamine biosynthesis, and the Shannon index, while *Blautia* had a positive correlation with Shannon index, L-trytophan, and dTDP-N-acetylthomosamine (dTDP-4-amino-4,6-dideoxy-alpha-D-galactose-1) biosynthesis, and IgA. A positive correlation was observed between *Eggerthella*, *Anaerostipes*, *Ruminococcus gnavus* group, *Ruminococcus torques* group, *Flavonifractor*, and UBA1819, while a negative correlation was seen between *Veillonella* and the Shannon index. Furthermore, a positive correlation was found between *Akkermansia* and dTDP-N-acetylthomosamine biosynthesis, while a positive correlation was observed between *Clostridium sensu stricto* 1 and L-trytophan biosynthesis, IgA, and propionic acid (Appendix A). 

In the BF group, GI symptoms were positively associated with *Bacteroidetes*, *Blautia*, L-trytophan, and dTDP-N-acetylthomosamine biosynthesis at 6 months after the intervention. A negative correlation was found between *Bifidobacterium* and the Shannon index, species richness, NAD biosynthesis, and propionic acid. However, a positive correlation was found between this organism and lactic acid concentration. L-tryptophan and dTDP-N-acetylthomosamine biosynthesis were positively associated with *Blautia*. *Collinsella*, *Anaerostipes*, *Ruminococcus gnavus* group, *Ruminococcus torques* group, and *Flavonifractor* showed positive correlations with the Shannon index. In addition, *Flavonifractor* was positively correlated with the biosynthesis of NAD and L-tryptophan (Appendix A).

### 2.7. Major Bacteria Metabolic Pathways

When we evaluated the effects of INN formula treatment in infants, we observed that the important bacteria metabolic pathways were different compared to the STD group at 21 days, 2 months, and 6 months of life. Furthermore, the abundance of each metabolic pathway of the INN group was more similar to the BF group. At 21 days and 6 months, we observed that the bacterial NAD biosynthesis pathway was significantly lower in the INN group compared to the STD group. The catechol degradation pathway was significantly lower in the INN and BDF groups compared to the STD group at 21 days. At 2 months, the abundance of the octane oxidation pathway was significantly decreased in INN compared to STD, and the BF group exhibited values similar to the INN group. Furthermore, the (S)-propane-1,2-diol degradation pathway was found to be decreased in the INN and BF groups compared to the STD group at 2 and 6 months. At 6 months, we also found that the abundance of DTDP-N-acetylthomosamine biosynthesis and L-tryptophan biosynthesis pathways was significantly decreased in the INN compared to the STD group, and the BF group exhibited values similar to the INN group (Figure 4).

## 3. Discussion

The present study aimed to evaluate whether a novel starting formula (INN) versus a standard formula (STD) provides a gut microbiota composition more similar to that of breastfed infants (BF) for the first 6 months of life. Here, we show that the gut microbiota of infants consuming the INN formula was closer to that of those who were exclusively breastfed in terms of richness and diversity. This was true for *Bacteroides*, *Bifidobacterium*, *Clostridium*, and *Lactobacillus* at the genus level, and for calprotectin and SCFAs levels at 21 days, 2 months, and 6 months. As a result, we observed that the major bacteria metabolic pathways were more similar for the INN and BF groups compared with the STD group. These results indicate that consuming the starting novel INN formula could improve gut microbiota composition towards a healthier intestinal microbiota in breastfed infants.

### 3.1. Effects on Richness and Diversity

Previous studies have already demonstrated that breastfeeding causes less diversity in the gut microbiome compared to those who were given formula [24,25]. This indicates that the gut microbiome depends on the type of food consumed. Indeed, when we evaluated the richness and diversity in both formula and BF groups, we found that those infants who were exclusively breastfed exhibited lower levels of richness and diversity at 21 days. At 2 and 6 months, INN-formula-fed infants presented a diversity more similar to that obtained in BF infants, indicating that the INN formula may potentially point to an effect closer to that promoted by breastfeeding, with potential effects on metabolic and immune health. At the phylum level, the relative abundance was similar among groups at 21 days, 2 months, and 6 months. Nonetheless, the relative abundance of *Proteobacteria* was lower per treatment and per visit between BF and both formulas INN and STD, as was reported previously in 4-week-old Korean infants fed either human milk or formula [26]. Here, infants receiving the STD formula exhibited an increase in Shannon index, inverse Simpson, Pielou’s evenness, and Simpson at 21 days, 2 months, and 6 months; Fisher index and species richness were different per visit, with the highest relative abundance at 6 months. This is consistent with a previous meta-analysis which reported the effects of exclusive breastfeeding on infant gut microbiota across populations. In the first 6 months of life, gut bacterial diversity and the relative abundance of *Bacteroidetes* and *Firmicutes*, as we observed here, were consistently lower in breastfed infants compared to infants who were fed with formula or non-exclusive human milk [27]. Moreover, several studies have identified varying differences in gut microbial composition or diversity between exclusively breastfeeding (EBF) and non-EBF infants [28,29,30,31]. However, the fold change of absolute counts between visit 1 (21 days) and visit 4 (6 months) revealed differences between the STD group and the INN and BF groups, especially in the phylum *Verrucomicrobia*, and especially in the genera *Streptococcus, Ruminococcus gnavus* group, *Clostridium, Faecalibacterium, Flavonifractor*, and *Akkermansia*, revealing how STD feeding differs from the rest of the patterns.

### 3.2. Bifidobacterium and Other Genera

*Bifidobacterium* is a normal inhabitant of the intestine of healthy infants and adults. Its absence is related to the appearance of colic in infants [32]. Therefore, it has widely described beneficial functions, many of them associated with the prevention and treatment of colic intestinal diseases and immunological disorders [33]. Furthermore, a reduction in their abundance in infants increases the prevalence of obesity, diabetes, metabolic disorders, and all-cause mortality later in life [34]. A recent study detecting gut microbiota in infants fed exclusively human milk or a certain kind of formula for more than 4 months after birth showed that levels of *Bifidobacterium* and *Bacteroides* were significantly greater, while *Streptococcus* and *Enterococcus* were significantly lower in the breastfed group than in the formula-fed group [31]. Here, we revealed, at the genus level, that the relative abundance of *Bifidobacterium* was lower in the STD group compared with the INN and BF groups at 21 days, 2 months, and 6 months. Furthermore, the fold change of absolute counts between visit 1 (21 days) and visit 4 (6 months) revealed the highest value for *Bifidobacterium* in the INN group compared to the STD and BF groups, indicating a shift toward a healthier gut microbiota composition because this genus dominates the gut microbiota of breastfed infants at 12 months of age [35]. Indeed, we found that *Bifidobacterium* levels were negatively correlated with the Shannon index and propionic acid levels in the INN group. This is in agreement with previous reports in early life showing that increased *Bifidobacterium* is associated with lower alpha diversity as measured by the Shannon index [36]. In addition, an increase in the *Bifidobacterium* species, which is enriched in breastfed infants, is negatively associated with the fecal concentrations of propionic acid [37]. Beyond the beneficial effects in terms of plasma amino acid pattern, which is more similar to that of breastfed infants [38], the INN formula group might have exhibited a higher content of *Bifidobacterium* strains due to the higher α-lactalbumin content compared to the STD formula. Indeed, the growth-promoting properties of α-lactalbumin on several *Bifidobacterium* strains have been previously described, suggesting that it could modify the gut microbiota of formula-fed infants towards a pattern more similar to that of breastfed infants [39,40]. On the other hand, *Bacteroides* also exhibited differences, with the relative abundance of INN more similar to that of the BF group, indicating that the INN formula might mimic a gut microbiota composition closer to that promoted by human milk. The *Bacteroides* genus is related to greater intestinal diversity and the maturation of the gut microbiome, regardless of the mode of birth [41]. A recent study demonstrated that the *Bacteroides*-dominant gut microbiome of late infancy is associated with enhanced neurodevelopment at 1 and 2 years of age [42].

*Lactobacillus* is one of the first beneficial bacteria to colonize the intestinal tract of infants. Maternal microorganisms are considered to be the key source of bacteria during the development of gut microbiota in infants [43]. Indeed, a lack of *Lactobacillus*, along with that of *Bifidobacterium*, could lead to future morbidities related to allergies and asthma, among others [44]. The genera *Lactobacillus*, as well as *Staphylococcus*, presented higher relative abundances in the BF group, followed by the INN group and the lowest levels in the STD group.

Interestingly, the *Veillonella* genus is a minor component of bacteria taxa of the core infant gut that is saccharolytic and utilizes products of carbohydrate fermentation (e.g., lactate) of other infant gut bacteria, such *as Streptococcus* spp. and *Bifidobacterium* spp., to produce propionate, forming an influential trophic chain [45]. Indeed, we observed that lactic acid levels were higher in the BF and INN groups compared to the STD at 2 and 6 months of life. This could be related to *Bifidobacterium*, and it may be a source of *Veillonella* in the intestine. Furthermore, we found a positive correlation between *Bifidobacterium* and lactic acid, and a negative association with propionic acid and Shannon index in the BF group. This is in line with previous studies showing how human milk increases *Bifidobacterium* and lactic acid [37,46]. We observed the opposite effects for *Clostridium sensu stricto 1*. In the present study, the STD group presented the highest relative abundance at 21 days and 2 months compared to the INN and BDF groups. This indicates that infants fed the STD formula were potentially at a higher risk of infectious diseases, sepsis, and upper respiratory infection [47]. Similarly, *C. difficile* levels were lower in the INN and BF groups, which can be interpreted as a reduced risk factor for infectious diarrhea in infants [48,49]. Other genera such as *Collinsella* and *Akkermansia* revealed differences per visit, showing a lower relative abundance in the STD group compared with the INN and BF groups. In the case of *Collinsella*, it is a commensal with the ability to produce butyrate, one of the most bioactive SCFAs in controlling inflammatory responses [50]. *Akkermansia*, belonging to the phylum *Verrucomicrobia*, has been reported to be highly enriched in breastfed infants compared to formula-fed groups [51], and it has been associated with atopy and the development of asthma [52]. We also found a positive correlation between *Akkermansia* and the L-tryptophan and dTDP-N-acetylthomosamine biosynthesis pathways in the INN group. Interestingly, the STD group showed a negative association between the bacteria’s L-tryptophan biosynthesis pathway and *Bifidobacterium*, which was not observed in the INN and BF groups. Clinical studies investigate the involvement of tryptophan metabolites in the generation of microbiota–gut–brain axis signaling underlying major gut disorders such as irritable bowel and inflammatory bowel disease, both characterized by psychiatric disorders. Moreover, there is the possibility that L-tryptophan may be metabolized by the gut microbiota and exert direct or indirect control over its metabolism, giving rise to some compounds, such as 5-HT, kynurenines, tryptamine, and indolic compounds, which are involved in microbiota–gut–brain communication [53]. Therefore, modulating L-tryptophan metabolism by changing the composition of the microbiota might be a useful therapeutic approach.

In our study, we found differences in terms of modulation of the microbiome between the two evaluated formulas. This revealed that infants consuming the INN formula presented greater similarities, in terms of diversity and microbiome content, to the BF group. In fact, oligosaccharide content impacts the growth of some species, such as *B. longum*, *C. perfringens*, or *E. coli* [54]. Here, the oligosaccharide content was at the same level between both formulas; however, we should note the differentiated DHA levels. The INN formula contained an equal amount of DHA and AA, 24 mg of each per 100 kcal. This was compared to 10 mg of each fatty acid per 100 kcal in the STD formula. AA and DHA are associated with the genus *Bacteroides, Enterobacteriaceae, Veillonella, Streptococcus*, and *Clostridium*, bacteria involved in SCFA production. These bacteria have significant immunomodulatory functions and play a key role in the development of intestinal pathologies, among other functions. They significantly increased at 13–15 days after breastfeeding was initiated [55,56].

Recently, it was found that infant formula supplemented with *Lactobacillus paracasei* F19 decreased the diversity of gut microbiota compared to the standard group without the probiotic. This was similar to breastfeeding after 4 months. In addition, *L. paracasei* F19 increased lactobacilli and tended to increase Bifidobacteria. The most dominant genus in the infant microbiome was *Bifidobacterium* throughout the study, which persisted until the first year, making this more similar to breastfeeding [57]. In this context, Bazanella et al. reported that infants exposed to bifidobacteria-enriched formula exhibited decreased amounts of *Bacteroides* and *Blautia* spp. associated with changes in lipids at one month. This is because the supplementation of Bifidobacteria to the infant diet can modulate the occurrence of specific bacteria and metabolites during early life [58]. According to another study, a formula containing the probiotic *Bifidobacterium lactis* may improve the composition of the gut microbiota in low-birth-weight infants. This may increase the *Bifidobacterium* and *Lactobacillus* genera while decreasing the *Veillonella*, *Dolosigranulum*, and *Clostridium* genera [59]. Infant formula supplemented with bovine-milk-derived oligosaccharides and *Bifidobacterium animalis* subsp. *lactis* CNCM I-3446 showed a shift to bifidobacteria increasing the *B. lactis* by 100-fold in the stool, and other species such as *B. longum, B. breve, B. bifidum,* and *B. pseudocatenulatum* [60].

In the present study, the INN formula contained thermally inactivated postbiotic *Bifidobacterium animalis* subsp. *lactis*, which might confer some benefits concerning body composition, metabolism, and gut microbiota composition. It is well known that probiotics and postbiotics have health benefits by modulating the gut microbiome. *Bifidobacterium animalis* subsp. *lactis* reduced total lipid and triacylglycerols in the nematode *C. elegans* [18], increasing survival and modulating tryptophan metabolism; it also reduced the ratio of plasma cholesterol total/LDL-cholesterol in obese rats [18], and reduced body mass index (BMI), waist circumference, and visceral fat in individuals with abdominal obesity after twelve-week treatment [19]. Interestingly, postbiotic *Bifidobacterium animalis* subsp. *lactis* increased in *Akkermansia* spp., particularly in the live form, which was inversely related to body weight. Here, we also observed a higher relative abundance of *Akkermansia* in infants fed with the INN formula compared to those fed the STD, making it more similar to breastfeeding. Remarkably, we showed a positive association between *Akkermansia* and the bacteria L-tryptophan biosynthesis pathway in the INN group, which indicates that the postbiotic might alter tryptophan metabolism.

### 3.3. Secretory IgA

Secretory IgA is essential for the immune defense system of the intestinal mucosa in the first years of life. We found that it increased in the BF group; however, no differences were found between the INN and STD groups. IgA plays a key role in the immune exclusion of pathogens and the development of oral tolerance to commensal intestinal bacteria [61]. The fecal IgA values of children fed by breastfeeding decrease after 6 months. However, the rates remain significantly higher for children in the INN and STD groups. The INN group tends to present values more similar to breastfeeding, although not statistically significant. This trend at 6 months of age could represent an advantage of the INN formula, as it provides better mucosal defense in these formula-fed infants; it has been shown that secretory IgA levels in formula-fed infants are lower and have a delayed acquisition time compared to breastfed-infants in their first year of life [62]. We also observed a positive correlation between IgA levels and lactic acid in the INN group, which showed a higher *Bifidobacterium* level compared to the STD group. In the case of fecal calprotectin, we did not find differences between the INN and BF groups at 21 days. This calcium-binding protein, produced by neutrophils, granulocytes, and macrophages in the submucosa, has been described as a valuable marker of intestinal inflammation, intestinal diseases, including inflammatory bowel disease, and neoplasms, and the possible filtration of the intestinal barrier [63].

### 3.4. Short-Chain Fatty Acids

Regarding the effects on SCFAs, there was a significant increase in the levels of lactic acid in the BF and INN groups compared to the STD. This was at 2 and 6 months of life. Lactic acid is associated with lower pH levels. This is a better environment for the growth of beneficial bacteria such as bifidobacteria [64], the main protective factor against gastrointestinal infections. Indeed, we also found that bifidobacteria was greater in the BF group, followed by those fed the INN formula. Infants fed with both formulas showed higher levels of butyrate at 21 days compared with those in the BF group. Nevertheless, butyrate levels were significantly higher in the STD group at 2 and 6 months. This is in line with a previous study showing that butyrate was lower in formulas supplemented with human milk oligosaccharides (HMG), and in the breastfeeding group compared to the standard formula group [48], indicating a more diverse microbiota in the standard group, as we observed here. Butyrate, indeed, is produced by *Bacteroides* and *Firmicutes* (e.g., *Clostridium*), but not by *Bifidobacterium* [65].

### 3.5. Metabolic Pathways Profile

Beyond taxonomic composition, we conducted functional profiling of gut microbiota to identify the bacteria metabolic pathways involved in the effects of consuming the INN and STD formulas, also compared to BF, at 21 days, 2 months, and 6 months. Interestingly, we observed that bacteria metabolic pathways were different in the INN and BF groups compared to the STD group. At 21 days, the abundance of the microbial NAD biosynthesis pathway was significantly lower in the INN group compared to the STD group, indicating that the INN formula might decrease bacteria contributing to mammalian host NAD biosynthesis through a microbial nicotinamidase [66]. At 6 months, the NAD biosynthesis pathway displayed the same pattern. Similarly, the catechol degradation pathway was higher in the STD group compared to both the INN and BDF groups. Catechol is an intermediate in the degradation of many different aromatic compounds, and it is important in the bacteria of many genera, including species of *Azotobacter*, *Ralstonia*, and numerous species of *Pseudomonas*. Hence, at 21 days, the INN formula showed a bacteria metabolic profile closer to the BF group. At 2 months, we observed that the abundance of the octane oxidation pathway was significantly decreased in the INN group compared to STD, and the BF group exhibited values similar to the INN group, which supports a closer bacteria metabolic profile to breast milk in the INN formula. The octane oxidation pathway involves the alkane hydroxylase system that introduces molecular oxygen in the C1 atom of the hydrocarbons at the expense of NADH to yield primary alcohols [67]. Previous studies demonstrated that several alkane-degrading bacteria can use diverse compounds as a carbon source in addition to alkanes, which are further oxidized to fatty acids via the bacterial β-oxidation pathway [68]. The (S)-propane-1,2-diol degradation pathway was found to be decreased in the INN and BF groups compared to STD at 2 and 6 months. (S)-propane-1,2-diol (propylene glycol) is produced from (S)-lactaldehyde during the bacterial degradation of L-rhamnose and L-fucopyranose. *Salmonella enterica* can utilize (S)-propane-1,2-diol as a carbon source and its metabolism may be a virulence factor [69,70]; therefore, lower levels of this bacteria metabolic pathway might be beneficial for the host. At 6 months, we also found that abundance of DTDP-N-acetylthomosamine biosynthesis and L-tryptophan biosynthesis pathways was significantly decreased in the INN compared to the STD group, and the BF group exhibited values similar to the INN group. Indeed, INN formula contains higher whey proteins, particularly α-lactalbumin, which is relatively rich in tryptophan. This protein is rapidly digested and participates in the building of muscle mass, as it remains mostly soluble in the stomach and passes more rapidly toward the intestine. Hence, lower bacterial abundance of L-tryptophan biosynthesis pathway in INN and BF groups might be related, at least in part, to the higher availability of this in the intestine.

### 3.6. Strengths, Limitations, and Suggestions

The main strength of the present study is that it was designed as a randomized, multicenter, double-blind, parallel, and comparative clinical trial of equivalence of two starting infant formulas for infants, including very tight eligibility, inclusion, and exclusion criteria. Indeed, the study was designed under the hypothesis that the weight gain achieved by infants fed formula 1 (INN) would be equivalent to that observed in children fed with formula 2 or STD. Furthermore, a third unblinded group of breastfed infants was used as a further reference group for exploratory analysis.

However, the study has several limitations. The first is that the number of infants per group was not initially calculated to estimate a potential difference in microbial diversity or specific bacterial groups from the fecal microbiota but to evaluate possible differences in growth. Probably, to obtain significant differences in certain underrepresented bacterial groups, the number of infants analyzed in the present study would have to be higher. Another limitation is derived from the microbiota methodology. We used a methodology that involves the amplification of specific regions of the bacterial 16S ribosomal DNA, but it would be desirable to use the complete sequencing of the entire gene, which would allow a better diagnosis of bacterial species beyond families or genera. Another limitation of the study is that the variations observed in the fecal microbiota cannot be assigned to a specific component of the experimental formula, since its composition differs in several components in comparison to the standard milk formula.

Shortly, new longitudinal studies should be designed from birth to the start of weaning with statistically sufficient numbers of infants fed with milk formulas that differ in a single component compared to standard formulas, especially when nutrients, probiotic or postbiotic microorganisms known to have a strong influence on the intestinal microbiota, are included.

## 4. Materials and Methods

### 4.1. Ethics

This clinical trial was carried out following the recommendations of the International Conference on Harmonization Tripartite on good clinical practice, the ethical-legal principles established in the latest revision of the Declaration of Helsinki, as well as the current regional regulations that regulate pharmacovigilance and food safety. More information regarding the study protocol can be found in Ruiz-Ojeda et al., 2022 [71].

### 4.2. Trial Design

The INNOVA study was designed as a randomized, multicenter, double-blind, parallel, and comparative clinical trial of the equivalence of two starting infant formulas for infants. Furthermore, a third unblinded group of breastfed infants was used as a further reference group for exploratory analysis. Blinding for both investigator and participant remained assured as both infant formulas were labeled the same. It is not mandatory to carry out specific clinical tests to demonstrate the nutritional and healthy properties of infant formulas as per the current EU legislation (EC Regulation No. 1924/2006).

This study evaluated the safety, tolerance, effects on growth, incidence of major acute infectious diseases, and changes in gut microbiota for 6 months and up to 12 months after the introduction of complementary feeding (INNOVA study 2020). The primary objective of the study was to determine if the mean weight gain between treatment groups 1 and 2 was equivalent. The chosen primary endpoint of weight gain is recommended as the primary endpoint by the “American Academy of Pediatrics Guidelines” [72]. According to previous studies carried out in infants fed with different infant formulas from 0 to 6 months, the average weight gain with infant formula was around 20–25 grams/day with a standard deviation between 5 and 6 grams/day. A difference in mean weight gain of 3 grams/day will be considered clinically relevant in most of these studies. To resolve this contrast, we use a test for independent samples. With a power of 80%, a significance level of 5%, an equivalence cut-off of 3 g/day, and a common standard deviation of 5.5 g/day, we would need to recruit 59 children for each group. Furthermore, if the loss rate is 20%, it would be necessary to include 70 infants, that is, a total of 210 children (70 per group).

We did not find differences between groups in weight gain, BMI, body composition length, head circumference, and tricipital/subscapular skinfolds. Nevertheless, there were fewer respiratory, thoracic, and mediastinal disorders among BF children. In addition, infants receiving the INN formula experienced significantly fewer general disorders and disturbances than those receiving the STD formula. In fact, atopic dermatitis, bronchitis, and bronchiolitis were substantially more prevalent among infants fed the STD formula than those provided with the INN formula or BF [73].

The infants were selected by primary care pediatricians through active and consecutive recruitment. Pediatricians informed and invited parents of 15-day-old infants who visited their offices regularly (for regular medical check-ups) to be involved in the trial. Infants that were not candidates for breastfeeding (for different reasons) were proposed to participate in the formula-feeding groups. To keep the three arms of the trial balanced, one candidate breastfeeding subject was recruited at each center for every two infants supplemented with infant formula.

### 4.3. Study Groups

The study was carried out in 21 centers, all located in Spain, of which 17 recruited at least one subject. In total, 217 subjects signed the informed consent (IC) and 145 were randomized to receive one of the two infant formulas.

Group 1 (infant formula 1): Nutribén Innova® 1 (INN)

This formula had a lower amount of total protein and a proportion of casein to whey protein ratio of about 70/30 by increasing the content of α-lactalbumin compared to STD formula. Furthermore, the INN formula contained higher levels of both AA and DHA and a postbiotic (thermally inactivated *Bifidobacterium animalis* subsp. *lactis*, which is a trademark owned by ADM Biopolis S.L., Spain (CECT8145), registered in the European Union, the USA, and other countries).

Group 2 (infant formula 2): Nutribén® standard (STD)Group 3: Breastfeeding (external control exploratory analysis)

The experimental product object of this trial (INN) and the STD formula comply with the recommendations of the ESPGHAN (European Society of Pediatrics Gastroenterology, Hepatology, and Nutrition) and with Regulation 609/2013 of the European Parliament and of the Council regarding foods intended for children, infants and young children, foods for special medical purposes and complete diet substitutes for weight control and repealing Council Directives 92/51, Directives 96/8/EC, 1999/21/CE, 2006/125/CE, and 2006/141/CE of the Commission, Directive 2009/38/CE of the European Parliament and of the Council and Regulations 41/2009 and 953/2009 of the Commission. More detailed information on the composition of each of the products can be found in the additional information (Appendix A) [73]. Both formulas were given to infants ad libitum. The two trial formulations were administered following the preparation instructions in the manufacturer’s package insert. DHA was obtained from purified and concentrated fish oil (DSM Health, Nutrition & Bioscience, Basel, Switzerland).

### 4.4. Inclusion and Exclusion Criteria

The selection of the children of the breastfeeding group was carried out among those infants who met the inclusion and exclusion criteria of the study [71]. Participating infants should meet all inclusion criteria and exclusion criteria as follows: (1) healthy children of both sexes; (2) term children (between 37 and 42 weeks of gestation); (3) birth weight between 2500 g and 4500 g; (4) single delivery; and (5) mothers with a BMI, before pregnancy, between 19 and 30 kg/m^2^. Volunteers were excluded from participation based on the following criteria: (1) body weight less than the 5th percentile for that gestational age; (2) allergy to cow’s milk proteins and/or lactose; (3) history of antibiotic use during the 7 days before inclusion; (4) congenital disease or malformation that can affect growth; (5) diagnosis of disease or metabolic disorders; (6) significant prenatal and/or severe postnatal disease before enrollment; (7) minor parents (younger than 18 years old); (8) newborn of a diabetic mother; (9) newborn of a mother with drug dependence during pregnancy; (10) newborn whose parents/caregivers cannot comply with procedures of the study; (11) infants participating in or have participated in another clinical trial since their birth.

The experimental product object of this trial (INN) and the STD formula comply with the recommendations of the ESPGHAN (European Society of Pediatric Gastroenterology, Hepatology, and Nutrition) and with Regulation 609/2013 of the European Parliament and of the Council regarding foods intended for children, infants and young children, foods for special medical purposes and complete diet substitutes for weight control and repealing Council Directives 92/51, Directives 96/8/EC, 1999/21/CE, 2006/125/CE and 2006/141/CE of the Commission, Directive 2009/38/CE of the European Parliament and the Council and Regulations 41/2009 and 953/2009 of the Commission. More detailed information on the composition of each of the products can be found in the additional information (Appendix A) [73]. Both formulas were given to infants ad libitum. The two trial formulations were administered following the preparation instructions in the manufacturer’s package insert. DHA was obtained from purified and concentrated fish oil (DSM Health, Nutrition & Bioscience, Basel, Switzerland).

### 4.5. Sampling

Fecal samples were collected at 21 days, 2 and 6 months using a collection kit provided by ADM-Biopolis (Valencia, Spain), which included a sample-stabilizing buffer to ensure its stability. The samples were processed and sequenced according to the original codes provided by the researchers, assigning the group in the final bioinformatics analysis.

### 4.6. DNA Extraction

DNA extraction was carried out using a previously optimized protocol, which includes a combination of beads beating and enzymatic lysis, following a modified protocol from Yuan et al. [74] and applying the QIAmp Power Fecal Kit (Qiagen, Germany). The DNA quality control was performed using Nanodrop equipment (ThermoFisher, Madrid, Spain) to ensure the DNA had the minimum conditions for extraction. DNA yield was calculated by measuring absorbance ratios spectrophotometrically, including A260/230 nm for salt and phenol contamination and A260/280 nm for protein contamination.

### 4.7. Sequencing and Bioinformatic Analysis

The amplification of the extracted DNA was performed by PCR using the primers for 16S, targeting the V3 and V4 hypervariable regions of the bacterial 16S rRNA gene [75], marked with a molecular identifier and performing a primer dimer cleanup. The libraries were sequenced on Illumina’s Novaseq 6000 platform combined with 250PE (Illumina, Madrid, Spain). A negative control containing water was obtained to confirm the absence of contamination.

Illumina bcl2fastq2 Conversion Software v2.20 was used to demultiplex raw sequences, and raw data were imported into QIIME 2 2020.8 open-source software [76] using the q2-tools-import script which uses the PairedEndFastqManifestPhred33 input format. Denoising was performed using DADA2 [77], which uses a quality-aware model of Illumina amplicon errors to obtain a distribution of sequence variances, each differing by one nucleotide. To truncate the forward reads at position 288 and trim them at position 6, the q2-dada2-denoise script was executed following the retrieving quality scores. We trimmed reverse reads at position 7 after truncating them at position 220. To remove chimeras, we applied the “consensus” filter, which detects chimeras in samples individually and removes those found in a sufficient fraction of samples. Additionally, forward and reverse reads are merged during this step. Phylogenies were constructed with FASTTREE2 (via q2-phylogeny) [78] using all amplicon sequence variants (ASVs) aligned with MAFFT [79] via q2-alignment. To classify ASVs, a naïve Bayes taxonomy classifier was used (via q2-feature-classifier) [80] against the SILVA 16S V3-V4 v132_99 [81] along with a similarity threshold of 99%. As part of the data filtering process, samples with fewer than 10,000 reads were excluded.

The diversity of the samples was studied using the vegan library [82]. On the one hand, alpha diversity indices were studied, such as Shannon, Simpson and species richness, and Pielou’s evenness.

On the other hand, beta diversity was also studied using Bray–Curtis distances and multidimensional ordering techniques. PERMANOVA tests were performed to verify the significance of these results. A comparative study was carried out between the four periods of the study and the three proposed treatments. This study was developed using the DESeq2 tool, in which a negative binomial distribution is assumed in the count matrix to proceed with the test Wald statistics that allow us to discern whether there is a differential effect according to the time or treatment between the samples.

### 4.8. Functional Profiles

Potential functional profiles for sequenced samples were predicted using PICRUSt2 [83]. In summary, phylotypes were placed into a reference tree containing 20,000 full 16S rRNA genes from prokaryotic genomes in the Integrated Microbial Genomes (IMG) database. Functional annotation of these genomes was based on the Clusters of Orthologous Groups of proteins (COG) and the Enzyme Commission numbers (EC) databases. To obtain a deeper understanding of the biomolecular activity of the microbial communities, we conducted functional profiling of gut microbiota to identify the bacteria metabolic pathways involved in the effects of consuming the INN or STD formulas, compared also with BF at 21 days, 2 months, and 6 months. To infer MetaCyc pathways, EC numbers were first regrouped to MetaCyc reactions. Pathway abundances were calculated as the harmonic mean of the key reaction abundances in each sample. To infer the abundance of each gene family per sample, the abundances of phylotypes were corrected by their 16S rRNA gene copy number and then multiplied by their functional predictions.

### 4.9. Biochemical Analysis

Calprotectin and IgA levels were determined by ELISA kit according to the manufacturer’s instructions. Calprotectin levels are associated with inflammation [84], and secreted IgA is essential for the immune defense system of the intestinal mucosa in the first years of life [61]. Lactic acid and SCFAs (acetic, butyric, and propionic acids) regulate microbial homeostasis by maintaining an acidic milieu that inhibits colonization by pathogens [85]. The determinations were carried out by high-performance liquid chromatography (HPLC). An Alliance 2695 HPLC equipment coupled to a refractive index detector was used. The column used was an Aminex HPX-87H from Bio-Rad (Madrid, Spain, 300 mm × 7.8 mm) at a temperature of 60 °C. Isocratic elution was performed with 5 mM H_2_SO_4_ at a flow rate of 0.6 ml min^-1^. The identifications were performed by comparison with the retention time of the standards and calibration curves were used for the quantifications.

### 4.10. Rivera-Pinto Analysis

Rivera-Pinto analysis identifies microbial signatures, that is, groups of microbial taxa that are predictive of a phenotype of interest. These microbial signatures can be used for diagnosis, prognosis, or prediction of therapeutic response based on an individual’s specific microbiota. Hence, the identification of microbial signatures involves both modeling and variable selection, i.e., modeling the response variable and identifying the smallest number of taxa with the highest prediction or classification accuracy. Here, the Rivera-Pinto method and selbal algorithm, which is a model selection procedure that searches for a sparse model that adequately explains the response variable of interest, were used to assess specific signatures at the phylum and genus levels; this method considers microbial signatures generated by the geometric means of data from two groups of taxa whose relative abundances, or balances, are related to the response variable of interest [23].

### 4.11. Statistical Analysis

Bacterial data are expressed as median and range and diversity indices are expressed as mean ± standard error (SEM). To determine differences in phyla and genera in response to intervention time (visit) and treatment, a general linear model for repeated measures (GM) was used, which includes the analysis of treatment, visit, and the interaction visit × treatment. *p*-values were determined for time and treatment × time; different letters mean significant differences (*p* < 0.05) and were calculated with Least Significant Difference test (LSD) post hoc multiple comparisons for observed means.

Functional pathway profile data and SCFAs are given as the mean and SEM. *p* < 0.05 was considered to be statistically significant. Variables that were not normally distributed were log-transformed for analysis, and/or values with ±3SD of the mean (outliers) were removed (without achieving values loss from samples of up to 15%). However, the data are presented as untransformed values to ensure a clear understanding. For the relative abundances of bacteria (phylum and genus), the U Mann–Whitney test was applied for assessing differences at baseline, as well as for the alpha indexes and beta diversity. Statistical tests were performed using IBM SPSS Statistics for Windows, Version 25.0 (IBM Corp., Armonk, NY, USA).

All figures from metabolic pathways were assembled in GraphPad Prism 8 (GraphPad Software, San Diego, CA, USA, version 8.0.0). Data are presented as mean ± SEM unless stated differently in the figure legend. Statistical significance was determined by using one-way ANOVA, followed by Tukey’s multiple comparison test, or as stated in the respective figure legend. Differences reached statistical significance with *p* < 0.05. The relationships between the diversity indices, microbiome variables, metabolic parameters, SCFA levels, and clinical outcomes (bronchiolitis and gastrointestinal symptoms have been published elsewhere [23], as the presence or absence of these symptoms) were examined using Pearson’s correlations at 6 months of the intervention. Using the corrplot function in R studio software (R Foundation for Statistical Computing, Vienna, Austria), associations were expressed by correcting multiple testing with the FDR procedure [86]. Only significant and corrected associations are shown in the graph [87]. Red and blue lines indicate the correlation values within the graphs, with negative correlations shown in red (-1) and positive correlations shown in blue (+1).

## 5. Conclusions

Infants consuming the INN formula, compared to the STD formula, exhibited gut microbiota compositions closer to those infants that were breastfed in general terms of richness and diversity, presence of genera such as *Bifidobacterium*, *Lactobacillus*, *Bacteroides*, and *Clostridium*, calprotectin, and SCFA levels at 21 days, 2 months, and 6 months. Additionally, we observed that the major bacteria metabolic pathways between the INN formula and BF groups were more similar compared to the STD formula group. This indicates, henceforth, that consuming the novel INN formula may improve gut microbiota composition towards a healthier intestinal microbiota.

## Figures and Tables

**Figure 1 ijms-24-03034-f001:**
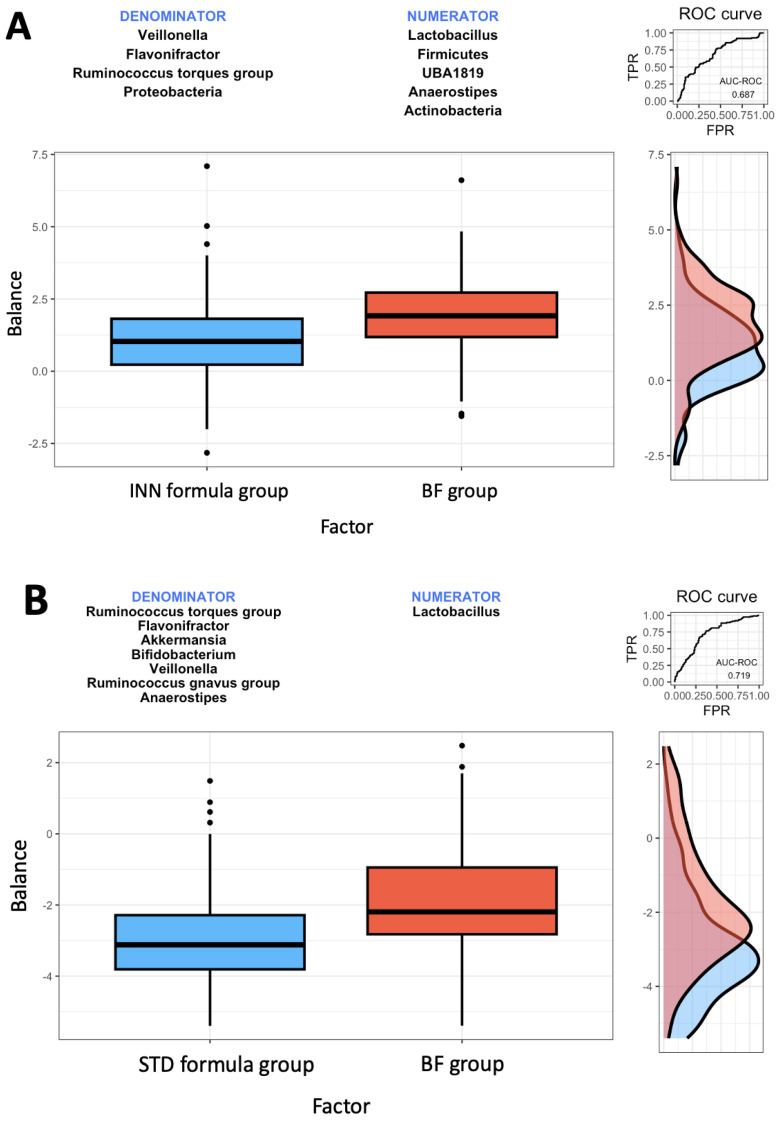
Description of the global balance for groups. The two groups of taxa that form the global balance are specified at the top of the plot. The box plot represents the distribution of the balance scores for INN and BF group (**A**), STD and BF groups (**B**), and INN and STD groups (**C**). The right part of the figure contains the ROC curve with its AUC (mean area under the ROC curve) value and the density curve for each group.

**Figure 2 ijms-24-03034-f002:**
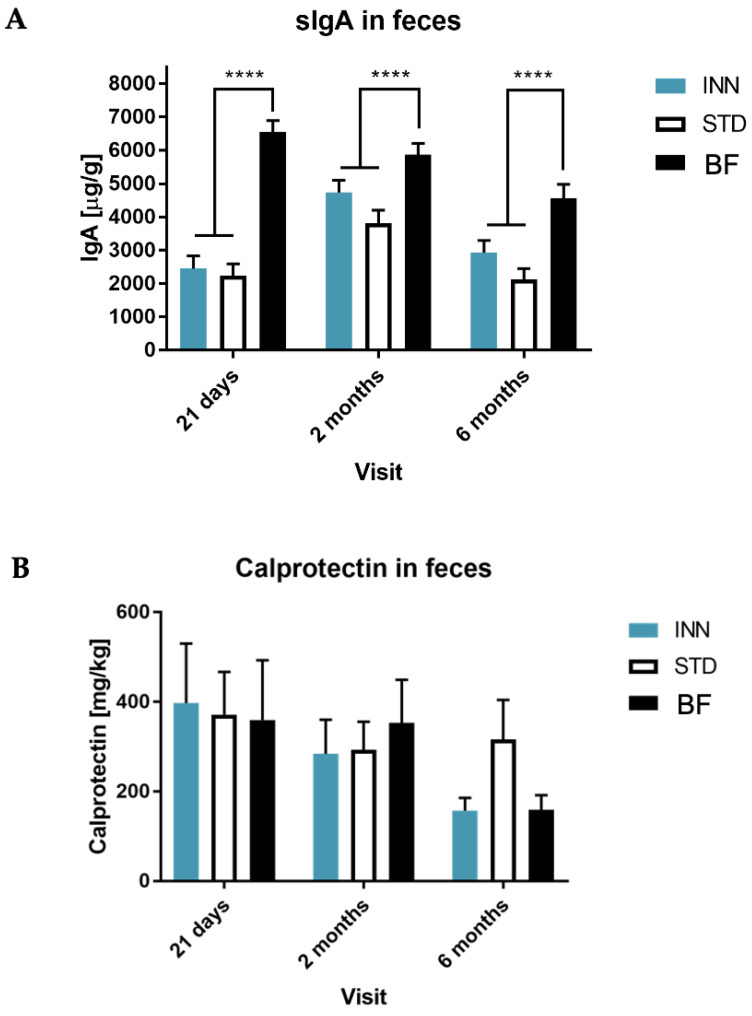
Secretory IgA and calprotectin in feces, including INNOVA formula (INN), standard formula (STD), and breastfeeding (BF) at 21 days, 2 months, and 6 months. (**A**) IgA; (**B**) Calprotectin. **** *p* < 0.0001.

**Figure 3 ijms-24-03034-f003:**
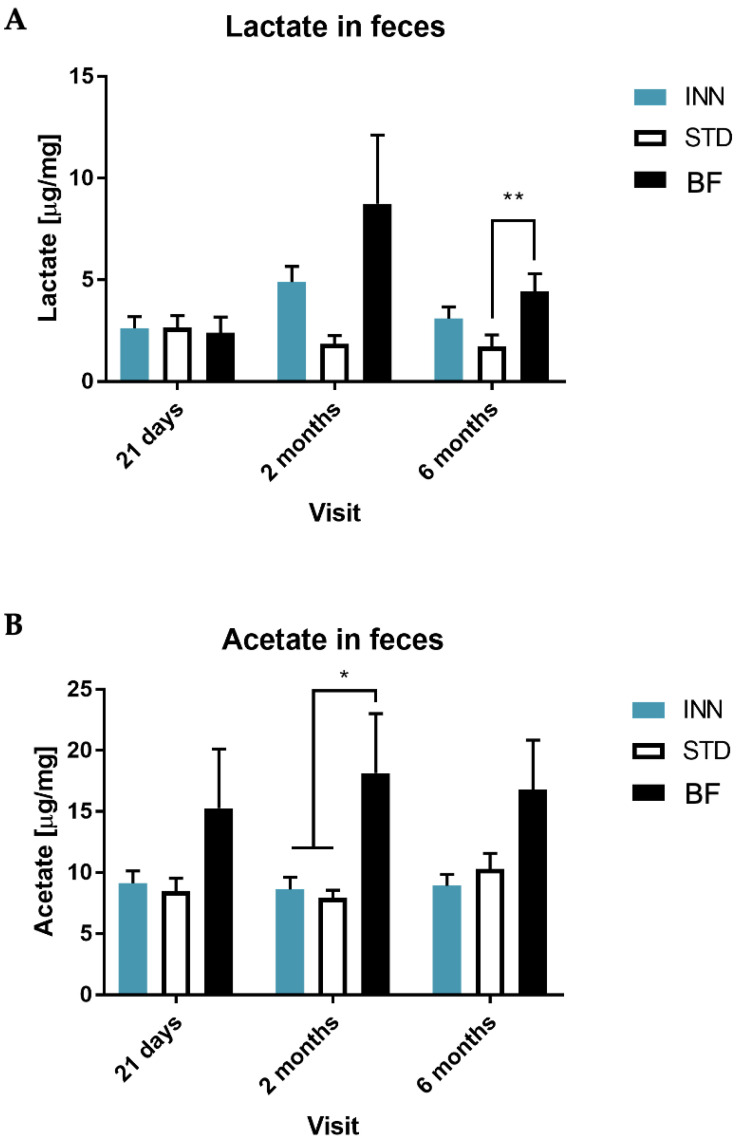
Lactate, acetate, propionate, and butyrate in feces, including INNOVA formula (INN), standard formula (STD), and breastfeeding (BF) at 21 days, 2 months, and 6 months. (**A**) Lactate, (**B**) Acetate, (**C**) Propionate, (**D**) Butyrate, * *p* < 0.05, ** *p* < 0.01, *** *p* < 0.001 and **** *p* < 0.001.

**Figure 4 ijms-24-03034-f004:**
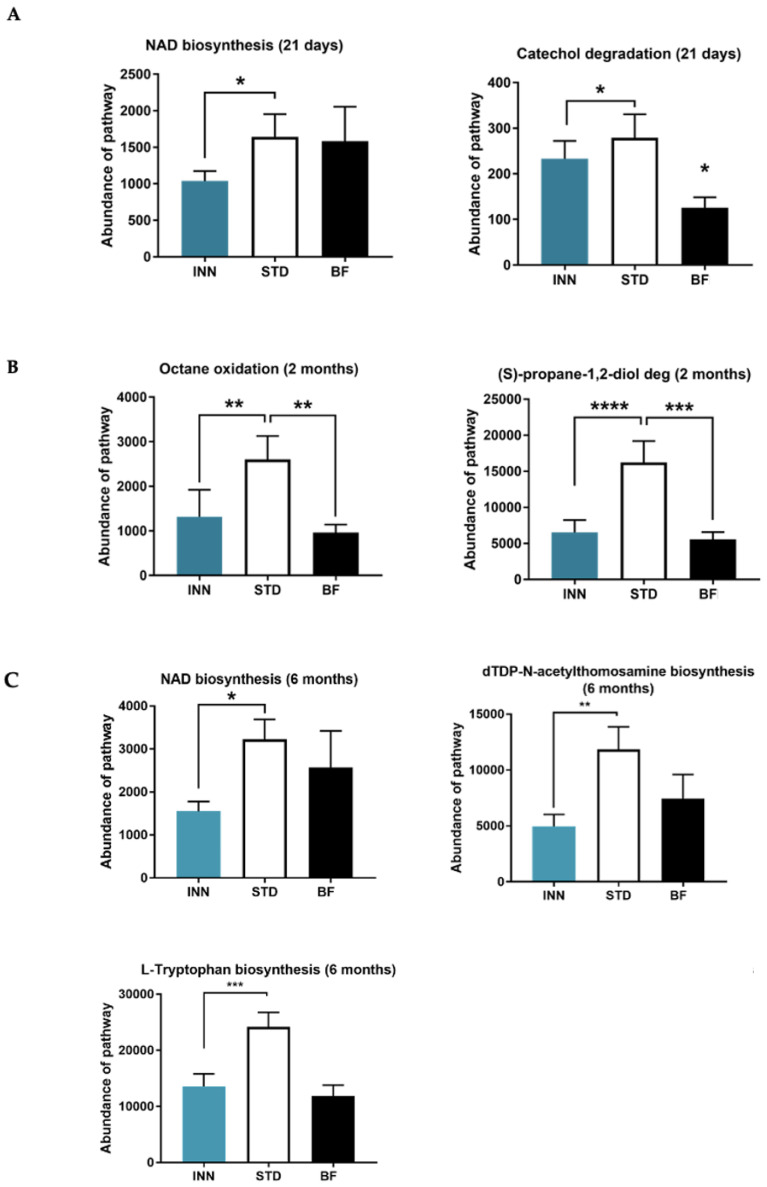
Major metabolic pathway categories relative to the abundance of each sample, including INNOVA formula (INN), standard formula (STD), and breastfeeding (BF). (**A**) NAD biosynthesis (21 days), and Catechol degradation (21 days), (**B**) Octane oxidation (2 months) and (S)-propane-1,2-diol deg (2 months), (**C**) NAD biosynthesis (6 months), dTDP-N-acetylthomosamine biosynthesis (6 months), L-Tryptophan biosynthesis, * *p* < 0.05, ** *p* < 0.01, *** *p* < 0.001 and **** *p* < 0.001.

**Table 1 ijms-24-03034-t001:** Relative abundances at the phylum levels of fecal bacteria in infants fed the INNOVA (INN) or a standard formula (STD) compared to exclusive breastfed (BF) infants up to 6 months of age.

Phylum	21 Days			2 Months			6 Months			*p*-Values		
	BF (n = 68)	STD (n = 70)	INN (n = 75)	BF (n = 64)	STD (n = 64)	INN (n = 63)	BF (n = 52)	STD (n = 55)	INN (n = 52)	Treatment	Visit	Treatment × Visit
Fisher	5.1 ± 0.3	6.2 ± 0.3	6.2 ± 0.3	5.5 ± 0.3	5.6 ± 0.3	5.6 ± 0.3	4.9 ± 0.4	5.4 ± 0.3	4.8 ± 0.3	0.083	0.015	0.235
Shannon index	0.8 ± 0.07	1.1 ± 0.07	1.2 ± 0.06	0.9 ± 0.07	1.1 ± 0.07	0.7 ± 0.07	0.9 ± 0.08	1.2 ± 0.08	0.8 ± 0.08	<0.001	0.082	<0.001
Inverse Simpson	1.9 ± 0.2	2.3 ± 0.2	2.4 ± 0.1	1.8 ± 0.2	2.4 ± 0.2	1.6 ± 0.2	1.9 ± 0.2	2.4 ± 0.2	1.6 ± 0.2	<0.001	0.061	0.004
Pielou’s evenness	0.23 ± 0.02	0.28 ± 0.02	0.31 ± 0.02	0.23 ± 0.02	0.31 ± 0.02	0.19 ± 0.02	0.25 ± 0.02	0.32 ± 0.02	0.22 ± 0.02	<0.001	0.108	<0.001
Species richness	37.7 ± 1.9	45.4 ± 1.9	44.9 ± 1.8	41.0 ± 2.0	41.4 ± 2.0	41.2 ± 2.0	37.2 ± 2.3	40.7 ± 2.1	36.4 ± 2.2	0.058	0.023	0.213
Simpson	0.33 ± 0.03	0.41 ± 0.03	0.47 ± 0.03	0.35 ± 0.03	0.47 ± 0.03	0.28 ± 0.03	0.37 ± 0.03	0.48 ± 0.03	0.32 ± 0.03	<0.001	0.251	<0.001
*Actinobacteria*	90.5 (1.2–99.4)	76.4 (2.9–99.4)	65.3 (3.2–98.9)	89.2 (1.2–99.2)	76.6 (2.7–98.9)	92.9 (10.8–99.1)	88.4 (4.3–99.1)	80.5 (1.5–98.6)	92.5 (6.0–98.0)	0.599	0.205	0.247
*Firmicutes*	8 (0.4–98.5)	21.6 (0.4–96.6)	32.9 (0.8–96.3)	10.1 (0.6–98.5)	19.6 (1.0–96.9)	5.8 (0.6–62.6)	10.2 (0.8–95.6)	15.8 (1.2–98.3)	6.9 (1.3–93.8)	0.838	0.168	0.235
*Verrucomicrobia*	0.08 (0–3.3)	0.1 (0–71.9)	0.08 (0–55.8)	0.06 (0–1.6)	0.06 (0–62.5)	0.06 (0–42.0)	0.05 (0–10.7)	0.05 (0–66.6)	0.04 (0–35.3)	0.198	0.641	0.625
*Proteobacteria*	0.22 (0–1.9)	0.4 (0.01–2.1)	0.4 (0–2.6)	0.2 (0–1.1)	0.4 (0.04–2.7)	0.5 (0.04–1.2)	0.2 (0–1.1)	0.2 (0.003–0.8)	0.3 (0.01–1.2)	0.008	<0.001	0.98
*Bacteroidetes*	0.04 (0–0.6)	0.02 (0–1.0)	0.04 (0–0.7)	0.04 (0–1.5)	0.07 (0–1.1)	0.03 (0–0.7)	0.02 (0–1.0)	0.1 (0–0.7)	0.08 (0–0.8)	0.828	0.242	0.261
*Fusobacteriota*	0 (0–1.0)	0 (0–4.9)	0 (0–0.6)	0 (0–1.9)	0 (0–0.1)	0 (0–0.3)	0.0008 (0–4.2)	0 (0–1.2)	0.004 (0–0.4)	0.58	0.407	0.538
*Patescibacteria*	0 (0–0.9)	0 (0–0.02)	0 (0–0.02)	0 (0–0.1)	0 (0–0.3)	0 (0–0.2)	0 (0–3.2)	0 (0–0.01)	0 (0–0)	0.558	0.076	0.931
*Synergistetes*	0 (0–0.3)	0 (0–0.3)	0 (0–0.09)	0 (0–0.3)	0 (0–0.07)	0 (0–0.04)	0 (0–0.05)	0 (0–0.3)	0 (0–0.2)	0.156	0.447	0.366
*Cyanobacteria*	0 (0–0.1)	0 (0–0.1)	0 (0–0.07)	0 (0–0)	0 (0–0.04)	0 (0–0.02)	0 (0–0.1)	0 (0–0.4)	0 (0–0.09)	0.599	0.677	0.902

Diversity indices are expressed as mean ± standard error, and phylum relative abundances are expressed as median and range. A general linear model for repeated measures was used to determine differences due to intervention time and treatment. *p*-values were determined for time and treatment × time; different letters mean significant differences (*p* < 0.05) and were calculated with Least Significant Difference test (LSD) post hoc multiple comparisons for observed means.

**Table 2 ijms-24-03034-t002:** Relative abundances at the genus levels of fecal bacteria in infants fed the INNOVA (INN) or a standard formula (STD) compared to exclusive breastfed (BF) infants up to 6 months of age.

Genus	21 Days			2 Months			6 Months			*p*-Values		
	BF (n = 68)	STD (n = 70)	INN (n = 75)	BF (n = 64)	STD (n = 64)	INN (n = 63)	BF (n = 52)	STD (n = 55)	INN (n = 52)	Treatment	Visit	Treatment × Visit
*Bifidobacterium*	79.9 (0.7–99.4)	70.1 (2.3–98.6)	50.3 (3.2–97.1)	70.4 (0.9–98.9)	67.2 (2.1–94.9)	87.4 (7.7–98.9)	78.7 (3.6–98.9)	68.5 (0.8–98.3)	82.7 (4.8–95.7)	0.002	<0.001	<0.001
*Clostridium sensu stricto 1*	1.1 (0–92.5)	1.8 (0.06–82.9)	5.8 (0.1–93.8)	1.0 (0–94.2)	2.5 (0.2–62.2)	1.2 (0.02–37.0)	0.5 (0–72.4)	0.5 (0–88.5)	1.4 (0.01–16.9)	0.104	<0.001	0.039
*Collinsella*	0.3 (0–30.1)	0.3 (0–33.4)	0.4 (0–44.0)	0.3 (0–88.6)	0.3 (0.01–23.8)	0.4 (0–53.4)	0.4 (0.03–50.9)	0.3 (0.01–54.9)	0.6 (0–63.7)	0.331	0.001	0.845
*Blautia*	0.08 (0–30.6)	0.1 (0–94.2)	0.1 (0–2.4)	0.1 (0–18.0)	0.09 (0–89.7)	0.06 (0–10.5)	0.06 (0–12.3)	0.1 (0–41.4)	0.05 (0–13.3)	0.016	0.973	1
*Ruminococcus gnavus group*	0.1 (0–95.6)	0.2 (0–43.3)	0.2 (0–15.0)	0.2 (0–90.7)	0.2 (0–73.9)	0.08 (0–10.1)	0.2 (0.01–41.0)	1.6 (0–46.7)	0.2 (0–73.9)	0.009	0.362	0.827
*Clostridioides*	0.08 (0–2.9)	0.05 (0–70.0)	0.1 (0–32.2)	0.04 (0–47.2)	0.07 (0–25.0)	0.03 (0–8.5)	0.04 (0–5.8)	0.3 (0–24.3)	0.03 (0–39.7)	0.086	0.967	0.535
*Akkermansia*	0.06 (0–3.3)	0.09 (0–70.9)	0.08 (0–55.2)	0.05 (0–1.6)	0.07 (0–62.0)	0.05 (0–42.4)	0.06 (0–10.9)	0.05 (0–66.1)	0.04 (0–35.6)	<0.001	0.472	0.848
*Eggerthella*	0.1 (0–18.2)	0.1 (0–43.5)	0.2 (0–35.2)	0.09 (0–6.8)	0.1 (0–38.4)	0.05 (0–23.8)	0.2 (0–5.2)	0.5 (0–15.5)	0.09 (0–5.7)	0.041	0.905	0.15
*Terrisporobacter*	0 (0–1.2)	0 (0–28.9)	0.01 (0–77.8)	0 (0–0.8)	0.01 (0–23.6)	0 (0–2.7)	0 (0–0.3)	0 (0–0.5)	0 (0–1.0)	0.32	0.273	0.268
*Flavonifractor*	0.03 (0–2.1)	0.05 (0–6.3)	0.06 (0–44.1)	0.04 (0–2.0)	0.1 (0–59.3)	0.03 (0–3.8)	0.04 (0–13.4)	0.2 (0–9.6)	0.03 (0–3.2)	0.059	0.465	0.002
*Cutibacterium*	0.05 (0–15.7)	0.03 (0–11.8)	0.02 (0–1.9)	0.03 (0–70.9)	0.01 (0–0.2)	0.02 (0–1.9)	0 (0–0.5)	0 (0–0.2)	0 (0–1.6)	0.021	0.238	0.452
*Subdoligranulum*	0.01 (0–13.6)	0.02 (0–7.4)	0.03 (0–51.8)	0.02 (0–19.9)	0.01 (0–0.8)	0 (0–0.8)	0.02 (0–19.5)	0.02 (0–13.6)	0 (0–0.2)	0.928	0.535	0.431
*Intestinibacter*	0 (0–0)	0 (0–0)	0 (0–0.1)	0 (0–0)	0 (0–0)	0 (0–0)	0 (0–0)	0 (0–0)	0 (0–0)	0.52	0.499	0.581
*Rothia*	0.06 (0–7.1)	0.1 (0–6.4)	0.1 (0–3.7)	0.08 (0–13.2)	0.09 (0–2.1)	0.1 (0–1.2)	0.04 (0–1.2)	0.02 (0–0.3)	0.06 (0–0.6)	0.087	0.002	0.129
*Faecalibacterium*	0 (0–7.3)	0.02 (0–5.5)	0.03 (0–23.3)	0.04 (0–1.6)	0 (0–1.8)	0.02 (0–8.8)	0.04 (0–5.5)	0.07 (0–27.1)	0.05 (0–3.6)	0.716	0.345	0.556
*Lachnoclostridium*	0 (0–7.2)	0.02 (0–15.9)	0.02 (0–18.0)	0.005 (0–0.9)	0.01 (0–15.7)	0 (0–3.1)	0 (0–17.3)	0.06 (0–4.1)	0 (0–3.5)	0.199	0.977	0.489
*Corynebacterium*	0 (0–0)	0 (0–0.3)	0 (0–0.2)	0 (0–0.2)	0 (0–0.3)	0 (0–0)	0 (0–0)	0 (0–0)	0 (0–0)	0.553	0.302	0.427
*UBA1819*	0 (0–8.6)	0 (0–0.8)	0 (0–24.6)	0 (0–6.4)	0 (0–5.8)	0 (0–3.3)	0 (0–12.5)	0 (0–16.6)	0 (0–2.5)	0.43	0.082	0.005
*Tyzzerella*	0 (0–0)	0 (0–0.2)	0 (0–0.1)	0 (0–0.3)	0 (0–0)	0 (0–0)	0 (0–0)	0 (0–0.2)	0 (0–0)	0.304	0.853	0.138
*Paeniclostridium*	0 (0–0.4)	0 (0–10.5)	0 (0–35.7)	0 (0–0.1)	0 (0–3.5)	0 (0–1.6)	0 (0–0.1)	0 (0–0.7)	0 (0–2.2)	0.458	0.257	0.686
*Peptoniphilus*	0 (0–22.2)	0.01 (0–1.6)	0.01 (0–7.9)	0 (0–0.3)	0.03 (0–2.0)	0.04 (0–1.1)	0 (0–3.5)	0.01 (0–0.1)	0 (0–0.6)	0.812	0.091	0.858
*Anaerostipes*	0.02 (0–4.0)	0.04 (0–16.8)	0.06 (0–1.9)	0.04 (0–5.4)	0.02 (0–23.2)	0.02 (0–7.4)	0.06 (0–6.2)	0.08 (0–6.6)	0.03 (0–3.0)	0.013	0.599	0.963
*Escherichia-Shigella*	0.1 (0–2.0)	0.2 (0–1.2)	0.1 (0–1.6)	0.01 (0–1.1)	0.3 (0–1.1)	0.3 (0–1.1)	0.2 (0–0.9)	0.1 (0–0.7)	0.3 (0–1.0)	0.012	0.608	0.06
*Anaerococcus*	0 (0–22.5)	0.02 (0–2.8)	0.02 (0–5.2)	0 (0–0.5)	0.02 (0–2.7)	0.03 (0–1.8)	0 (0–5.2)	0 (0–0.5)	0.01 (0–0.5)	0.715	0.125	0.871
*Finegoldia*	0 (0–5.8)	0.03 (0–1.6)	0.03 (0–7.8)	0.1 (0–0.4)	0.04 (0–4.7)	0.03 (0–4.2)	0 (0–5.1)	0.01 (0–0.1)	0.02 (0–0.4)	0.263	0.138	0.158
*Romboutsia*	0 (0–0.3)	0 (0–0.4)	0 (0–0.2)	0 (0–0)	0 (0–0.5)	0 (0–0)	0 (0–0)	0 (0–0.02)	0 (0–0.03)	0.246	0.406	0.881
*Streptococcus*	0.2 (0.01–1.9)	0.2 (0–1.1)	0.2 (0–1.1)	0.1 (0–1.4)	0.05 (0–0.6)	0.2 (0–1.2)	0.09 (0–1.4)	0.01 (0–0.9)	0.05 (0–0.7)	<0.001	<0.001	0.195
*Ruminococcus*	0 (0–5.3)	0 (0–5.3)	0 (0–0.4)	0 (0–1.1)	0 (0–1.8)	0 (0–0.3)	0 (0–0.2)	0 (0–0.7)	0 (0–0.2)	0.738	<0.001	0.706
*Eubacterium hallii group*	0 (0–2.4)	0 (0–1.6)	0.01 (0–22.8)	0 (0–0.6)	0 (0–29.5)	0 (0–0.5)	0 (0–9.7)	0.01 (0–6.0)	0.01 (0–1.2)	0.689	0.003	0.397
*Paraclostridium*	0 (0–0.3)	0 (0–15.3)	0 (0–9.2)	0 (0–0.9)	0 (0–1.3)	0 (0–20.8)	0 (0–0.1)	0 (0–0.1)	0 (0–1.2)	0.308	0.996	0.687
*Ruminococcus torques group*	0 (0–62.9)	0.03 (0–13.3)	0.03 (0–11.3)	0.04 (0–23.0)	0.01 (0–5.4)	0 (0–0.4)	0.01 (0–1.1)	0.03 (0–19.7)	0.01 (0–9.9)	0.548	0.47	0.534
*Veillonella*	0.03 (0–1.0)	0.05 (0–0.8)	0.2 (0–0.9)	0.03 (0–1.3)	0.2 (0–1.6)	0.1 (0–1.9)	0.1 (0–0.7)	0.1 (0–0.8)	0.2 (0–0.8)	<0.001	0.615	0.465
*Roseburia*	0 (0–0.4)	0 (0–0.5)	0 (0–0.3)	0 (0–0.3)	0 (0–0.1)	0 (0–0.5)	0 (0–0.07)	0 (0–0.09)	0 (0–0.08)	0.636	0.097	0.067
*Bacteroides*	0.04 (0–0.6)	0.03 (0–0.8)	0.04 (0–0.7)	0.06 (0–1.5)	0.06 (0–0.8)	0.02 (0–0.7)	0.01 (0–0.9)	0.07 (0–0.6)	0.1 (0–0.9)	0.944	0.023	0.075
*Enterococcus*	0.01 (0–0.6)	0.05 (0–1.4)	0.04 (0–1.2)	0.01 (0–0.5)	0.05 (0–0.5)	0.03 (0–0.7)	0.01 (0–0.4)	0.03 (0–1.0)	0.06 (0–0.8)	0.006	0.141	0.723
*Eubacterium*	0 (0–0.3)	0 (0–2.7)	0 (0–8.3)	0 (0–9.9)	0 (0–15.3)	0 (0–1.1)	0 (0–5.9)	0 (0–2.5)	0 (0–1.6)	0.705	0.379	0.573

Data are expressed as median and range. A general linear model for repeated measures was used to determine differences due to intervention time and treatment. *p*-values were determined for time and treatment × time; different letters mean significant differences (*p* < 0.05) and were calculated with Least Significant Difference test (LSD) post hoc multiple comparisons for observed means.

## Data Availability

The datasets used and/or analyzed during the current study are available from the corresponding author upon reasonable request.

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
