# Peer review of "Effects of a Novel Infant Formula on the Fecal Microbiota in the First Six Months of Life: The INNOVA 2020 Study"

_ijms, 2023, doi:10.3390/ijms24033034_

Round 1

Reviewer 1 Report

The study submitted to IJMS entitled “Effects of a Novel Infant Formula on the Fecal Microbiota of Infants: A Six-Month Follow-up of the INNOVA 2020 Study” aimed to evaluate the changes in the fecal microbiota and associated metabolic pathways during the first 6 months of life in infants fed a noel infant formula (INN), with a lower amount of protein, a proportion of casein to whey protein ratio of about 70/30 by increasing the content of α-lactalbumin, and with double the amount of docosahexaenoic acid/arachidonic acid compared with those fed the STD formula and a reference group of exclusively breastfed infants. The authors revealed how consuming the INN formula drives toward a more similar gut microbiota composition than those infants that were breastfed in general terms of richness and diversity, genera such as Bacteroides, Bifidobacterium, Clostridium and Lactobacillus, fecal secreted IgA, calprotectin, and SCFAs levels at 21 days, 2 and 6 months. The results and conclusions are supported by the data.

Minor comments:

1-Line 133: Please, modify the erratum “visit” for “visits”

2-Line 695: Please, modify the erratum “similar to infants breastfed” for “similar to infant’s breastfed”

3-What source of DHA was used?

Author Response

The study submitted to IJMS entitled “Effects of a Novel Infant Formula on the Fecal Microbiota of Infants: A Six-Month Follow-up of the INNOVA 2020 Study” aimed to evaluate the changes in the fecal microbiota and associated metabolic pathways during the first 6 months of life in infants fed a noel infant formula (INN), with a lower amount of protein, a proportion of casein to whey protein ratio of about 70/30 by increasing the content of α-lactalbumin, and with double the amount of docosahexaenoic acid/arachidonic acid compared with those fed the STD formula and a reference group of exclusively breastfed infants. The authors revealed how consuming the INN formula drives toward a more similar gut microbiota composition than those infants that were breastfed in general terms of richness and diversity, genera such as Bacteroides, Bifidobacterium, Clostridium and Lactobacillus, fecal secreted IgA, calprotectin, and SCFAs levels at 21 days, 2 and 6 months. The results and conclusions are supported by the data.

We would like to thank to the reviewer for the positive comments he/she made about our manuscript.

Line 133: Please, modify the erratum “visit” for “visits”

Response: The erratum has been modified.

Comment #2

Line 695: Please, modify the erratum “similar to infants breastfed” for “similar to infant’s breastfed”

Response: The erratum has been modified.

Comment #3

What source of DHA was used?

Response: DHA was isolated by using concentrate and purified fish oil (DSM Health, Nutrition & Bioscience, Basel, Switzerland). This has been included in the Methods section 4.3.

Reviewer 2 Report

Even though this study provides some interesting results, which may be relevant, the main objective of the study is not clear, the study design must have enough detail to be able to be replicated. The statistical significance of the results should be reassessed, the discussion should be improved. Some points need clarification, refinement, reanalysis, rewriting, and more information to improve this article. All these points must be resolved in the manuscript before it is considered for publication.

Major points

1.       The manuscript needs writing and language editing. The title should be improved, for example, “Effects of novel infant formula on the fecal microbiota in the first six months, INNOVA 2020 study”. What is the main aim of this study, apart from getting a novel formula with gets a good gain of weight in infants whose mothers cannot breastfeed (or decided not to do)? Is the main objective to analyse if the new formula against the standard provides a microbiome like that of breastfed children? The main objective must be direct and the same throughout the manuscript (abstract, introduction, results/discussion). The first time an abbreviation appears, the full name must be entered, for example, SCFAs: Short-chain fatty acids. What does BFD mean? Authors should not use the words that appear in the title as keywords. Line 103-106: Avoid repeating the same information. References should be recent and relevant, they should be well referenced, and their use should be improved throughout the manuscript.

2.       The introduction section should improve. An adequate presentation of the case under study must be made. Line 66: Add examples on this topic. What does US FDA mean? Lines 64-69: What is the purpose of this information for the main objective of this experimental study? Lines 97-103: This information and references must be written in the M&M section in a specific subsection on the physical, chemical, organoleptic characteristics, etc. of this novel formula. Lines 108-116: As this study had other targets whose results had been published, this information should appear in the M&M section in the features section of this novel formula. The research question should be clearly outlined. A good and clear justification for conducting this study should be given. It would be better if the authors offered a hypothesis before the main objective of this study. This could be the hypothesis: Children fed in the first 6 months with this new formula, compared to the standard formula, should develop a microbiota as similar as possible to the microbiome developed in breastfed children (control). What was the main objective of this study? It should be clear what the purpose of this study was.

3.       The materials and methods section is sparse and needs to be improved. The description must be clear, concise, and detailed. In a subsection, describe the main features of this new formula. Did this formula use any Bifidobacterium or similar bacteria? What do PCBs mean? Has reference 22 been published? Lines 780-802: Improve this description. It would be better to point out the inclusion and exclusion criteria. Authors must specify that this information has already been published (reference 23). When were the samples collected? What was the purpose of studying these parameters (IgA, calprotectin, lactic acid and SCFAs), and what would they reflect? How the authors calculated the sample size. How did the authors select the study population, by convenience sample? The authors should make a detailed description of the variables used in this study. All variables studied should be described (clinical outcome), defined, and measured appropriately. The description of the statistical analysis needs to be improved. What Statistical Program was used? This section should provide enough detail about the study design for it to be replicable.

4.       The results section: In the text, the authors should describe the most significant results. Authors should avoid repeating the same information in the text if it appears in Tables and Figures. Authors should describe in the M&M section that measures were analysed by treatment, visit, and the interaction visit x treatment. Rivera-Pinto's balance method should be described in the M&M section. Line 189: This information should be written in the M&M section. The authors should improve the size of the words in Figure 1. 2.6: Where are these associations significant? How were the metabolic pathways of bacteria studied? This information should be written in the M&M section. What information do these metabolic pathways provide? It would be good to show in a figure how the evolution of the different variables studied by groups was. It should be clear what were the most significant results of this study.

5.       The discussion should improve and be more argumentative. This section should start with the main objective of this study and the most significant results. It would be a good idea to discuss point by point in several subsections. The results collected by the authors should be discussed from multiple angles and placed in context without overinterpreting them. The main objective of this study should be the same throughout the manuscript. Lines 516-522: It would be better to use this information in the introduction section to support the main objective. Lines 526-532: Avoid repeating the same information. Authors should focus on the main objective and the most relevant results. Lines 533-535: The authors said: "Here, we show that the gut microbiota of infants who consumed INN formula was more similar to that of those who were breastfed in terms of richness and diversity, Bacteroides, Bifidobacterium, Clostridium, and Lactobacillus, at the genus level, values of IgA secreted by feces, calprotectin and SCFA levels at 21 days, 2 and 6 months". Is this statement correct? Because the authors in the results section report that “The fecal-secreted IgA values were higher for the BFD group for the entire duration of the study, compared to the two formula-fed infant groups.” Lines 667-696: This new formula was not supplemented by any bacteria, right? Why is this information important? The authors can suggest that their new formula can be improved with this supplementation, in the suggestions paragraph. Lines 729-730: This information must be described in the M&M section. All discussed results of the studied variables should have been described in the M&M section. A paragraph of limitations and suggestions for this study should be written before the conclusion.

6.       The conclusion should be like the summary. The introduction, the study design, and the discussion of the results should lead the reader to the same conclusion as the authors.

I encourage the authors to rewrite the manuscript, thinking about the principal goal of this study, and its design and answering with the results and arguments of the discussion the most proper conclusion to this research work.

Author Response

Response: We would like to thank the reviewer for the comprehensive revision of our manuscript. Following his/her suggestions:

-Title has been corrected to “Effects of a Novel Infant Formula on the Fecal Microbiota in the First Six Months of Life: the INNOVA 2020 study”.

-We have included the hypothesis of the study (Page 3) “We hypothesized that children fed in the first 6 months of life with a novel starting infant formula (Nutribén Innova® 1 or INN), compared to those fed a standard formula (Nutriben® or standard (STD)), should develop a microbiota as similar as possible to the microbiome developed in breastfed children (control or BF).”

-We have modified the objective of the study in the abstract, introduction and discussion sections.

Abstract: “Here, we aimed to evaluate whether a novel starting formula versus a standard formula provides a gut microbiota composition more similar to that of breastfed infants in the first 6 months of life.”

Introduction: The INN formula contains a lower amount of protein, a proportion of casein to whey protein ratio of about 70/30 by increasing the content of α-lactalbumin, and with double the amount of DHA/AA than the standard formula; it also contains a postbiotic, a thermally inactivated bacteria (Bifidobacterium animalis subsp. lactis, BPL1TM HT). Therefore, we evaluated whether the novel starting formula against a standard formula provides a gut microbiota composition similar to that of breastfed infants in the first 6 months of life.”

Discussion: “The present study aimed to evaluate whether a novel starting formula (INN) versus a standard formula (STD) provides a gut microbiota composition more similar to that of breastfed infants (BF) for the first 6 months of life.”

-SCFAs: Short-chain fatty acids have been included.

-BFD mean? BFD has now been changed to BF and it means “Breastfed” group, which has been included throughout the manuscript.

-Line 103-106: Avoid repeating the same information. Now, we have removed and modified the introduction section

-References should be recent and relevant, they should be well referenced, and their use should be improved throughout the manuscript. We have updated some relevant references.

-The manuscript has now been revised by a native English person.

Comment #2

The introduction section should improve. An adequate presentation of the case under study must be made. Line 66: Add examples on this topic. What does US FDA mean? Lines 64-69: What is the purpose of this information for the main objective of this experimental study? Lines 97-103: This information and references must be written in the M&M section in a specific subsection on the physical, chemical, organoleptic characteristics, etc. of this novel formula. Lines 108-116: As this study had other targets whose results had been published, this information should appear in the M&M section in the features section of this novel formula. The research question should be clearly outlined. A good and clear justification for conducting this study should be given. It would be better if the authors offered a hypothesis before the main objective of this study. This could be the hypothesis: Children fed in the first 6 months with this new formula, compared to the standard formula, should develop a microbiota as similar as possible to the microbiome developed in breastfed children (control). What was the main objective of this study? It should be clear what the purpose of this study was.

Response: As the reviewer suggested, we have modified:

            -Introduction section. We have included the hypothesis and objectives.

-Line 66: Add examples on this topic. What does US FDA mean? U.S. Food and Drug Administration This information has been modified in the Introduction section.

-Lines 64-69: What is the purpose of this information for the main objective of this experimental study? This information has been deleted.

-Lines 97-103: This information and references must be written in the M&M section in a specific subsection on the physical, chemical, organoleptic characteristics, etc. of this novel formula. We have included this information in the M&M section: “This formula had a lower amount of total protein and a proportion of casein to whey protein ratio of about 70/30 by increasing the content of α-lactalbumin. Furthermore, the INN formula contained higher levels of both AA and DHA, and a postbiotic (thermally inactivated Bifidobacterium animalis subsp. lactis, BPL1TM HT, which is a trademark owned by ADM Biopolis S.L., Spain, registered in the European Union, the USA and other countries).”

-Lines 108-116: As this study had other targets whose results had been published, this information should appear in the M&M section in the features section of this novel formula. This information has been modified accordingly: “In this study, the safety, tolerance, and effects on growth and incidence of major acute infectious diseases and changes in intestinal microbiota were evaluated for 6 months, and until 12 months after the introduction of complementary feeding (INNOVA 2020 study). We did not find differences in weight gain, BMI, body composition length, head circumference and tricipital/subscapular skinfolds between groups. Nevertheless, there were fewer respiratory, thoracic, and mediastinal disorders among BF children. In addition, infants receiving the INN formula experienced significantly fewer general disorders and disturbances than those receiving the STD formula. In fact, atopic dermatitis, bronchitis, and bronchiolitis were significantly more prevalent among infants who were fed STD formula compared to those fed BF formula or INN formula [77].

Comment #3

The materials and methods section is sparse and needs to be improved. The description must be clear, concise, and detailed. In a subsection, describe the main features of this new formula. Did this formula use any Bifidobacterium or similar bacteria? What do PCBs mean? Has reference 22 been published? Lines 780-802: Improve this description. It would be better to point out the inclusion and exclusion criteria. Authors must specify that this information has already been published (reference 23). When were the samples collected? What was the purpose of studying these parameters (IgA, calprotectin, lactic acid and SCFAs), and what would they reflect? How the authors calculated the sample size. How did the authors select the study population, by convenience sample? The authors should make a detailed description of the variables used in this study. All variables studied should be described (clinical outcome), defined, and measured appropriately. The description of the statistical analysis needs to be improved. What Statistical Program was used? This section should provide enough detail about the study design for it to be replicable.

Response: Following the reviewer’s suggestions and comments, we have included:

-In the subsection “4.3 Formula characteristics”, we have included the following information: “This formula had a lower amount of total protein and a proportion of casein to whey protein ratio of about 70/30 by increasing the content of α-lactalbumin. Furthermore, the INN formula contained higher levels of both AA and DHA, and a postbiotic (thermally inactivated Bifidobacterium animalis subsp. lactis, BPL1TM HT, which is a trademark owned by ADM Biopolis S.L., Spain, registered in the European Union, the USA and other countries).”

-Did this formula use any Bifidobacterium or similar bacteria? Yes, the INN formula contains a postbiotic, a thermally inactivated bacteria (Bifidobacterium animalis subsp. lactis, BPL1TM HT). This information is published in our protocol and through the present manuscript.

-What do PCBs mean? We apologize for the mistake as the acronym was in Spanish (Buenas prácticas Clínicas): In English is Good clinical practice. This erratum has been corrected.

-Has reference 22 been published? Yes, this reference is now online published in Advances in Pediatric Research journal: “Ruiz-Ojeda, F.J., et al., A multicenter, randomized, blinded, controlled clinical trial investigating the effect of a novel infant formula on the body composition of infants: INNOVA 2020 study protocol. Advances in Pediatric Research, 2022. 9(5).” This is now included in the references section.

-Lines 780-802: Improve this description. Using the reviewer’s comment, these lines were modified and now state “In this study, the safety, tolerance, and effects on growth and incidence of major acute infectious diseases and changes in intestinal microbiota were evaluated for 6 months, and until 12 months after the introduction of complementary feeding (INNOVA 2020 study). We did not find differences in weight gain, BMI, body composition length, head circumference and tricipital/subscapular skinfolds between groups. Nevertheless, there were fewer respiratory, thoracic, and mediastinal disorders among BF children. In addition, infants receiving the INN formula experienced significantly fewer general disorders and disturbances than those receiving the STD formula. In fact, atopic dermatitis, bronchitis, and bronchiolitis were significantly more prevalent among infants who were fed STD formula compared to those fed BF formula or INN formula [77].”

-It would be better to point out the inclusion and exclusion criteria. This has been included: “Inclusion criteria: participating infants should meet all inclusion criteria in the absence of exclusion criteria. Inclusion criteria were: 1) Healthy children of both sexes; 2) Term children (between 37 and 42 weeks of gestation); 3) Birth weight between 2500 g- 4500 g; 4) Single delivery; and 5) Mothers with a BMI, before pregnancy, between 19 and 30 kg/m2. Exclusion criteria: Volunteers were excluded from participation based on the following criteria: 1) Body weight less than the 5th percentile for that gestational age; 2) Allergy to cow's milk proteins and/or lactose; 3) History of antibiotic use during the 7 days before inclusion; 4) Congenital disease or malformation that can affect growth; 5) Diagnosis of disease or metabolic disorders; 6) Significant prenatal and/or severe postnatal disease before enrollment; 7) Minor parents (younger than 18 years old); 8) Newborn of a diabetic mother; 9) Newborn of a mother with drug dependence during pregnancy; 10) Newborn whose parents/caregivers cannot comply with procedures of the study; 11) Infants participating or have participated in another clinical trial since their birth.”

-Authors must specify that this information has already been published (reference 23): This reference was published online the 28 Dec 2022 after the initial submission of the present manuscript (24 Dec 22). Now, it is online (PMID: 36615804). We have included it in the references section.

-When were the samples collected? Fecal samples were collected at 21 days, 2 and 6 months. We have included this information in the manuscript.

-What was the purpose of studying these parameters (IgA, calprotectin, lactic acid and SCFAs), and what would they reflect? We have included this information in the M&M section: “Calprotectin levels are associated with inflammation [89], and secreted IgA is essential for the immune defense system of the intestinal mucosa in the first years of life [67]. Lactic acid and SCFAs (acetic, butyric and propionic acids) regulate microbial homeostasis by maintaining an acidic milieu that inhibits colonization by pathogens [90].”

-How the authors calculated the sample size.

Thanks to the reviewer for his/her comment, this information was added in the M&M section (4.2) and now states “A sample size of 210 children (70 per group) was determined based on the primary variable weight gain (the variable chosen). It is recommended that the primary endpoint be weight gain (as recommended by the "Guidelines from the American Academy of Pediatrics") [78]. According to previous studies conducted on infants fed different infant formula formulations from 0 to 6 months, the average weight gain with infant formula was around 20-25 grams/day with a standard deviation between 5 and 6 grams/day. A difference in mean weight gain of 3 grams/day was considered clinically relevant in most of these studies.

The primary objective of the study was to determine whether the mean weight gain between treatment groups 1 and 2 was equivalent. To resolve this contrast, we used a t-test for independent samples to resolve this contrast. With a power of 80%, a significance level of 5%, an equivalence limit of 3 g/day, and a common standard deviation of 5.5 g/day, we would need to recruit 59 children for each group. It will be necessary to include 70 infants in each of the groups, i.e., a total of 140 infants, if the loss rate is 20%.

Additionally, a third group of the same size (70 infants) was included in secondary comparisons between the formula-feeding and breast-feeding groups, which maintained the same significance. As with the main comparison, this secondary comparison maintained the same significance and power.”

-How did the authors select the study population, by convenience sample? Thanks to the reviewer for his/her comment, this information was added in the M&M section (4.2) and now states “The infants were selected by primary care pediatricians through active and consecutive recruitment; i.e., pediatricians informed and invited parents of 15-day-old infants not being breastfed for some reason to be involved in the trial (see details below under item 4.3). The study was carried out in 21 centers, all located in Spain, of which 17 recruited at least one subject. In total, 217 subjects signed the informed consent (IC) and 145 were randomized to receive one of the two infant formulas.”

-The authors should make a detailed description of the variables used in this study. All variables studied should be described (clinical outcome), defined, and measured appropriately. We are grateful for the reviewer's comment. The variables used in the present study are detailed in sections 4.3 to 4.8, and the only clinical outcomes are described elsewhere (reference 23).

-The description of the statistical analysis needs to be improved. What Statistical Program was used? This section should provide enough detail about the study design for it to be replicable. Thanks to the reviewer for his/her comment, the programs were added in the description and the section was modified and now states “4.10. Statistical analysis.

Bacterial data are expressed as median and range and diversity indices are expressed as mean ± standard error (SEM). To determine differences in phyla and genera in response to intervention time and treatment, a general linear model for repeated measures was used. P-values were determined for time and treatment x time; different letters mean significant differences (P<0.05) and were calculated with LSD post hoc multiple com-parisons for observed means.

Functional pathways profile data and SCFAs are given as the mean and SEM. P<0.05 was considered to be statistically significant. Variables that were not normally distributed were log-transformed for analysis, and/or values with ± 3SD of the mean (outliers) were removed (without achieving values loss from samples of up to 15%). However, the data are presented as untransformed values to ensure a clear understanding. For the relative abundances of bacteria (phylum and genus), the U Mann-Whitney test was applied for assessing differences at baseline, as well as for the alpha indexes and beta diversity. Statistical tests were performed using IBM SPSS Statistics for Windows, Version 25.0 (IBM Corp., Armonk, NY, USA).

All figures from metabolic pathways were ensemble GraphPad Prism 8 (GraphPad Software, San Diego, CA, USA, version 8.0.0). Data are presented as mean ± SEM unless stated differently in the figure legend. Statistical significance was determined by using one-way ANOVA, followed by Tukey’s multiple comparison test, or as stated in the respective figure legend. Differences reached statistical significance with p<0.05. The relationships between the diversity indices, microbiome variables, metabolic parameters, SCFA levels, and clinical outcomes (Bronchiolitis and gastrointestinal symptoms have been published elsewhere [23], as the presence or absence of these symptoms) were examined using Pearson's correlations at 6 months of the intervention. Using the corrplot function in R studio software (R Foundation for Statistical Computing, Vienna, Austria), associations were expressed by correcting multiple testing with the FDR procedure [91]. Only significant and corrected associations are shown in the graph [92]. Red and blue lines indicate the correlation values within the graphs, with negative correlations shown in red (-1) and positive correlations shown in blue (+1).

Comment #4

The results section: In the text, the authors should describe the most significant results. Authors should avoid repeating the same information in the text if it appears in Tables and Figures. Authors should describe in the M&M section that measures were analysed by treatment, visit, and the interaction visit x treatment. Rivera-Pinto's balance method should be described in the M&M section. Line 189: This information should be written in the M&M section. The authors should improve the size of the words in Figure 1. 2.6: Where are these associations significant? How were the metabolic pathways of bacteria studied? This information should be written in the M&M section. What information do these metabolic pathways provide? It would be good to show in a figure how the evolution of the different variables studied by groups was. It should be clear what were the most significant results of this study.

Response: As the reviewer suggested, we have included the following information:

-Authors should avoid repeating the same information in the text if it appears in Tables and Figures. Using the reviewer’s comments, repetition was reduced.

-Authors should describe in the M&M section that measures were analysed by treatment, visit, and the interaction visit x treatment. Using the reviewer’s comment, the information was added in the M&M section, “Bacterial data are expressed as median and range and diversity indices are expressed as mean ± standard error (SEM). To determine differences in phyla and genera in response to intervention time and treatment, a general linear model for repeated measures was used. P-values were determined for time and treatment x time; different letters mean significant differences (P<0.05) and were calculated with LSD post hoc multiple com-parisons for observed means.

-Rivera-Pinto's balance method should be described in the M&M section. We have included the following information in the manuscript: “4.9. Rivera-Pinto analysis. Rivera-Pinto analysis identifies microbial signatures, that is, groups of microbial taxa that are predictive of a phenotype of interest. These microbial signatures can be used for diagnosis, prognosis, or prediction of therapeutic response based on an individual’s specific microbiota. Hence, the identification of microbial signatures involves both modeling and variable selection: modeling the response variable and identifying the smallest number of taxa with the highest prediction or classification accuracy. Here, the Rivera-Pinto method and selbal algorithm, which is a model selection procedure that searches for a sparse model that adequately explains the response variable of interest, were used to assess specific signatures at the phylum and genus levels; this method considers microbial signatures generated by the geometric means of data from two groups of taxa whose relative abundances, or balances, are related to the response variable of interest [28].”

-Line 189: This information should be written in the M&M section. Mean area under the ROC curve (AUC) information has been included in the M&M section.

The authors should improve the size of the words in Figure 1. 2.6: Where are these associations significant? We have improved the quality of Figure 1.

How were the metabolic pathways of bacteria studied? This information should be written in the M&M section. According to the reviewer's comment, the section 4.7 has been modified and now states, “Potential functional profiles for sequenced samples were predicted using PICRUSt2 [88]. In summary, phylotypes were placed into a reference tree containing 20,000 full 16S rRNA genes from prokaryotic genomes in the Integrated Microbial Genomes (IMG) database. Functional annotation of these genomes was based on Clusters of Orthologous Groups of proteins (COG) and the Enzyme Commission numbers (EC) databases. To get a deeper understanding of the biomolecular activity of the microbial communities, we conducted functional profiling of gut microbiota to identify bacteria metabolic pathways involved in the effects of consuming INN or STD formulas, compared also with BF at 21 days, 2 and 6 months. To infer MetaCyc pathways, EC numbers were first regrouped to MetaCyc reactions. Pathway abundances were calculated as the harmonic mean of the key reaction abundances in each sample. To infer the abundance of each gene family per sample: the abundances of phylotypes were corrected by their 16S rRNA gene copy number and then multiplied by their functional predictions.

What information do these metabolic pathways provide? It would be good to show in a figure how the evolution of the different variables studied by groups was. It should be clear what were the most significant results of this study. We have included this information: “To get a deeper understanding of the biomolecular activity of the microbial communities, we conducted functional profiling of gut microbiota to identify bacteria metabolic pathways involved in the effects of consuming INN or STD formulas, compared also with BF at 21 days, 2 and 6 months.”

Comment #5

The discussion should improve and be more argumentative. This section should start with the main objective of this study and the most significant results. It would be a good idea to discuss point by point in several subsections. The results collected by the authors should be discussed from multiple angles and placed in context without overinterpreting them. The main objective of this study should be the same throughout the manuscript. Lines 516-522: It would be better to use this information in the introduction section to support the main objective. Lines 526-532: Avoid repeating the same information. Authors should focus on the main objective and the most relevant results. Lines 533-535: The authors said: "Here, we show that the gut microbiota of infants who consumed INN formula was more similar to that of those who were breastfed in terms of richness and diversity, Bacteroides, Bifidobacterium, Clostridium, and Lactobacillus, at the genus level, values of IgA secreted by feces, calprotectin and SCFA levels at 21 days, 2 and 6 months". Is this statement correct? Because the authors in the results section report that “The fecal-secreted IgA values were higher for the BF group for the entire duration of the study, compared to the two formula-fed infant groups.” Lines 667-696: This new formula was not supplemented by any bacteria, right? Why is this information important? The authors can suggest that their new formula can be improved with this supplementation, in the suggestions paragraph. Lines 729-730: This information must be described in the M&M section. All discussed results of the studied variables should have been described in the M&M section. A paragraph of limitations and suggestions for this study should be written before the conclusion.

Response: As the reviewer suggested, we have modified the discussion section.

-We have discussed point by point in several subsections and modified the discussion section.

-The main objective of this study has been included throughout the manuscript.

-Lines 516-522: It would be better to use this information in the introduction section to support the main objective. This information has been moved to the introduction section.

-Lines 526-532: Avoid repeating the same information. This has been deleted.

-Lines 533-535: The authors said: "Here, we show that the gut microbiota of infants who consumed INN formula was more similar to that of those who were breastfed in terms of richness and diversity, Bacteroides, Bifidobacterium, Clostridium, and Lactobacillus, at the genus level, values of IgA secreted by feces, calprotectin and SCFA levels at 21 days, 2 and 6 months". Is this statement correct? Because the authors in the results section report that “The fecal-secreted IgA values were higher for the BF group for the entire duration of the study, compared to the two formula-fed infant groups.” We apologize for this mistake. Indeed, we found that the fecal-secreted IgA values were higher for the BF group for the entire duration of the study, compared to the two formula-fed infant groups. We have modified the discussion section.

-Lines 667-696: This new formula was not supplemented by any bacteria, right? Why is this information important? The authors can suggest that their new formula can be improved with this supplementation, in the suggestions paragraph. We apologize if the information was not clear. In fact, this new formula was supplemented with a postbiotic, a thermally inactivated bacteria (Bifidobacterium animalis subsp. lactis, BPL1TM HT).

-Lines 729-730: This information must be described in the M&M section. All discussed results of the studied variables should have been described in the M&M section. A paragraph of limitations and suggestions for this study should be written before the conclusion. We have included new information in the M&M: “To get a deeper understanding of the biomolecular activity of the microbial communities, we conducted functional profiling of gut microbiota to identify bacteria metabolic pathways involved in the effects of consuming INN or STD formulas, compared also with BF at 21 days, 2 and 6 months.

Comment #6

The conclusion should be like the summary. The introduction, the study design, and the discussion of the results should lead the reader to the same conclusion as the authors. I encourage the authors to rewrite the manuscript, thinking about the principal goal of this study, and its design and answering with the results and arguments of the discussion the most proper conclusion to this research work.

Response: The conclusion section now is modified: “Infants consuming the INN formula, compared to STD formula, exhibited gut microbiota composition closer than those infants that were breastfed in general terms of richness and diversity, genera such as Bifidobacterium, Lactobacillus, Bacteroides, and Clostridium, calprotectin, and SCFAs levels at 21 days, 2 and 6 months. Additionally, we observed that the major bacteria metabolic pathways between the INN formula and BF groups were more similar compared to the STD formula group. This indicates, henceforth, that consuming the novel INN formula may improve gut microbiota composition towards a healthier intestinal microbiota.”

Round 2

Reviewer 2 Report

Even though the manuscript has been considerably improved, there are several points that require clarification, refinement, reanalysis, rewriting, and more information to improve this article before publication.

Line 40-43: It would be better to delete this paragraph, the information is not necessary.

Line 46: The conclusions should be the same.

Line 153: Authors should describe that measures were analysed by treatment, visit, and the interaction visit x treatment in the M&M section.

Lines 212-214: An AUC of 0.687 indicates poor discrimination accuracy between the INN formula group and the BF group.

Line 299: Make sure that only three subgroups can be seen.

Lines 308-309: It would be a good idea to write the full name and abbreviation in parentheses for all three groups.

Line 427: What does it mean?

Line 457: Although all the results are related, it would be better to separate them to give them more relevance. Could these topics be other subsections: Bifidobacterium, secretory IgA, calprotectin, SCFA, metabolic pathways, etc.? A paragraph of limitations and suggestions for this study should be written before the next subsection.

Lines 728-751: It would be a good idea to write these ideas as follows “This study evaluated the safety, tolerance, effects on growth, incidence of major acute infectious diseases, and changes in gut microbiota for 6 months and up to 12 months after the introduction of complementary feeding (INNOVA study 2020). The primary objective of the study was to determine if the mean weight gain between treatment groups 1 and 2 was equivalent. The chosen primary endpoint of weight gain is recommended as the primary endpoint by the "American Academy of Pediatrics Guidelines") [78]. According to previous studies carried out in infants fed with different infant formulas from 0 to 6 months, the average weight gain with infant formula was around 20-25 grams/day with a standard deviation between 5 and 6 grams/day. A difference in mean weight gain of 3 grams/day will be considered clinically relevant in most of these studies. To resolve this contrast, we use a test for independent samples. With a power of 80%, a significance level of 5%, an equivalence cut-off of 3 g/day, and a common standard deviation of 5.5 g/day, we would need to recruit 59 children for each group. Furthermore, if the loss rate is 20%, it would be necessary to include 70 infants, that is, a total of 210 children (70 per group). (In another paragraph): We did not find differences between groups in weight gain, BMI, body composition length, head circumference and tricipital/subscapular skinfolds. Nevertheless, there were fewer respiratory, thoracic, and mediastinal disorders among BFDBF children. In addition, infants receiving the INN formula experienced significantly fewer general disorders and disturbances than those receiving the STD formula. In fact, atopic dermatitis, bronchitis, and bronchiolitis were substantially more prevalent among infants fed the STD formula than those provided the INN formula or fed BF [77].”

Line 753-757: Pediatricians informed and invited parents of 15-day-old infants who visited their offices regularly (for regular medical check-ups) to be involved in the trial. Infants that were not a candidate for breastfeeding (for different reasons) were proposed to participate in the formula-feeding groups. To keep the three arms of the trial balanced, one candidate breastfeeding subject was recruited at each center for every two infants supplemented with infant formula.

Line 783: 4.3. Study groups / The study was carried out in 21 centers, all located in Spain, of which 17 recruited at least one subject. In total, 217 subjects signed the informed consent (IC) and 145 were randomized to receive one of the two infant formulas.

Line 789: Is there any code for this record?

Line 793: This paragraph should be written here: "The experimental product object of this trial (INN) and the STD formula comply with the recommendations of the ESPGHAN (European Society of Pediatrics Gastroenterology, Hepatology, and Nutrition) and with Regulation 609/2013 of the European Parliament and of the Council regarding foods intended for children, infants and young children, foods for special medical purposes and complete diet substitutes for weight control and repealing Council Directives 92/51, Directives 96/8/EC, 1999/21/CE, 2006/125/CE and 2006/141/CE of the Commission, Directive 2009/38/CE of the European Parliament and of the Council and Regulations 41/2009 and 953/2009 of the Commission. More detailed information on the composition of each of the products can be found in additional information [77] (Supplementary Table S1) [77]. Both formulas were given to infants ad libitum. The two trial formulations were administered following the preparation instructions in the manufacturer’s package insert. DHA was obtained from purified and concentrated fish oil (DSM Health, Nutrition & Bioscience, Basel, Switzerland)."

Lines 792-793: The infants were selected by primary care pediatricians through active and consecutive recruitment.

Line 799-813: In another subsection, such as 4.4. The inclusion and exclusion criteria.

Line 801: ...and exclusion criteria as follows: …

Line 805: The infants were: …

Line: 920: How many times were the variables measured, analyzed, and compared? at 21 days, 2 and 6 months? These data should be described in the Statistical Analysis in the M&M section. The authors should describe that measures were analysed by treatment, visit, and interaction visit x treatment in the M&M section. The mean area under the ROC curve (AUC) information has not been included in this section: The mean area under the ROC curve (AUC) less than 0.7 will indicate poor discrimination accuracy.

Author Response

Even though the manuscript has been considerably improved, there are several points that require clarification, refinement, reanalysis, rewriting, and more information to improve this article before publication.

Thank you again for the deep revision. Now, we have addressed all the comments in the manuscript.

Line 40-43: It would be better to delete this paragraph, the information is not necessary.

This has been removed

Line 46: The conclusions should be the same.

Now all the conclusions have been the same through the manuscript.

Line 153: Authors should describe that measures were analysed by treatment, visit, and the interaction visit x treatment in the M&M section.

We have described the measures by treatment, visit (time) and the interaction visit x treatment in the M&M section. To determine differences in phyla and genera in response to intervention treatment and time (visit), a general linear model for repeated measures (GM) was used, which it includes the analysis of treatment, visit, and the interaction treatment x visit. In fact, a GLM is a generalized model of variance for n way; in the present case two ways (treatment and time).

Lines 212-214: An AUC of 0.687 indicates poor discrimination accuracy between the INN formula group and the BF group.

This has been removed

Line 299: Make sure that only three subgroups can be seen.

Figure 1 represents the comparison of two groups: A: INN vs BF. B: STD vs BF, and C: INN vs STD.

Lines 308-309: It would be a good idea to write the full name and abbreviation in parentheses for all three groups.

The abbreviations have been included.

Line 427: What does it mean?

Least Significant Difference test. We have included this description in the manuscript.

Line 457: Although all the results are related, it would be better to separate them to give them more relevance. Could these topics be other subsections: Bifidobacterium, secretory IgA, calprotectin, SCFA, metabolic pathways, etc.? A paragraph of limitations and suggestions for this study should be written before the next subsection.

As the reviewer suggests, we have included the subsections in the Discussion section. Moreover, a general paragraph of limitations and suggestions was added at the end of the Discussion.

Lines 728-751: It would be a good idea to write these ideas as follows “This study evaluated the safety, tolerance, effects on growth, incidence of major acute infectious diseases, and changes in gut microbiota for 6 months and up to 12 months after the introduction of complementary feeding (INNOVA study 2020). The primary objective of the study was to determine if the mean weight gain between treatment groups 1 and 2 was equivalent. The chosen primary endpoint of weight gain is recommended as the primary endpoint by the "American Academy of Pediatrics Guidelines") [78]. According to previous studies carried out in infants fed with different infant formulas from 0 to 6 months, the average weight gain with infant formula was around 20-25 grams/day with a standard deviation between 5 and 6 grams/day. A difference in mean weight gain of 3 grams/day will be considered clinically relevant in most of these studies. To resolve this contrast, we use a test for independent samples. With a power of 80%, a significance level of 5%, an equivalence cut-off of 3 g/day, and a common standard deviation of 5.5 g/day, we would need to recruit 59 children for each group. Furthermore, if the loss rate is 20%, it would be necessary to include 70 infants, that is, a total of 210 children (70 per group). (In another paragraph): We did not find differences between groups in weight gain, BMI, body composition length, head circumference and tricipital/subscapular skinfolds. Nevertheless, there were fewer respiratory, thoracic, and mediastinal disorders among BF children. In addition, infants receiving the INN formula experienced significantly fewer general disorders and disturbances than those receiving the STD formula. In fact, atopic dermatitis, bronchitis, and bronchiolitis were substantially more prevalent among infants fed the STD formula than those provided the INN formula or fed BF [77].”

We have included this information in the M&M section.

Line 753-757: Pediatricians informed and invited parents of 15-day-old infants who visited their offices regularly (for regular medical check-ups) to be involved in the trial. Infants that were not a candidate for breastfeeding (for different reasons) were proposed to participate in the formula-feeding groups. To keep the three arms of the trial balanced, one candidate breastfeeding subject was recruited at each center for every two infants supplemented with infant formula.

This information has been included.

Line 783: 4.3. Study groups / The study was carried out in 21 centers, all located in Spain, of which 17 recruited at least one subject. In total, 217 subjects signed the informed consent (IC) and 145 were randomized to receive one of the two infant formulas.

This information has been included.

Line 789: Is there any code for this record?

We have included the code CECT8145 in the manuscript.

Line 793: This paragraph should be written here: "The experimental product object of this trial (INN) and the STD formula comply with the recommendations of the ESPGHAN (European Society of Pediatrics Gastroenterology, Hepatology, and Nutrition) and with Regulation 609/2013 of the European Parliament and of the Council regarding foods intended for children, infants and young children, foods for special medical purposes and complete diet substitutes for weight control and repealing Council Directives 92/51, Directives 96/8/EC, 1999/21/CE, 2006/125/CE and 2006/141/CE of the Commission, Directive 2009/38/CE of the European Parliament and of the Council and Regulations 41/2009 and 953/2009 of the Commission. More detailed information on the composition of each of the products can be found in additional information [77] (Supplementary Table S1) [77]. Both formulas were given to infants ad libitum. The two trial formulations were administered following the preparation instructions in the manufacturer’s package insert. DHA was obtained from purified and concentrated fish oil (DSM Health, Nutrition & Bioscience, Basel, Switzerland)."

We have included this information in the M&M section.

Lines 792-793: The infants were selected by primary care pediatricians through active and consecutive recruitment.

This has been corrected.

Line 799-813: In another subsection, such as 4.4. The inclusion and exclusion criteria.

This has been corrected.

Line 801: ...and exclusion criteria as follows: …

This has been corrected.

Line 805: The infants were: …

This has been corrected.

Line: 920: How many times were the variables measured, analyzed, and compared? at 21 days, 2 and 6 months? These data should be described in the Statistical Analysis in the M&M section. The authors should describe that measures were analysed by treatment, visit, and interaction visit x treatment in the M&M section. The mean area under the ROC curve (AUC) information has not been included in this section: The mean area under the ROC curve (AUC) less than 0.7 will indicate poor discrimination accuracy.

The variables have been measured, analyzed and compared at 21 days, 2 and 6 months. In fact, in the general linear model for repeated measures, such time x treatment effect information is included. These effects and interactions have been taken into account.
